# C3-OWD: A Curriculum Cross-modal Contrastive Learning Framework for Open-World Detection

## Abstract

Object detection has advanced significantly in the closed-set setting, but real-world deployment remains limited by two challenges: poor generalization to unseen categories and insufficient robustness under adverse conditions. Prior research has explored these issues separately: visible-infrared detection improves robustness but lacks generalization, while open-world detection leverages vision-language alignment strategy for category diversity but struggles under extreme environments. This trade-off leaves robustness and diversity difficult to achieve simultaneously. To mitigate these issues, we propose **C3-OWD**, a curriculum cross-modal contrastive learning framework that unifies both strengths. Stage 1 enhances robustness by pretraining with RGBT data, while Stage 2 improves generalization via vision-language alignment. To prevent catastrophic forgetting between two stages, we introduce an Exponential Moving Average (EMA) mechanism that theoretically guarantees preservation of pre-stage performance with bounded parameter lag and function consistency. Experiments on FLIR, OV-COCO, and OV-LVIS demonstrate the effectiveness of our approach: C3-OWD achieves $80.1$ $AP^{50}$ on FLIR, $48.6$ $AP^{50}_{Novel}$ on OV-COCO, and $35.7$ $mAP_r$ on OV-LVIS, establishing competitive performance across both robustness and diversity evaluations.

## 1 Introduction

Object detection (Zong et al., 2023; Zhu et al., 2020; Zhao et al., 2023; Lv et al., 2024) is a fundamental task in computer vision and has achieved remarkable progress under the closed-set setting, where models are trained and evaluated on a fixed set of categories.

Despite their efficiency and accuracy in standard benchmarks, traditional detectors face two key challenges when deployed in the real world:

- *limited generalization* to unseen categories
- *insufficient robustness* under adverse environmental conditions, as shown in Fig 1, such as low illumination, fog, etc.

To address these issues, two research directions have recently gained attention. On the one hand, visible–infrared object detection (RGBT-OD) (Shen et al., 2024b; Devaguptapu et al., 2019; Medeiros et al., 2024; Lee et al., 2024b) introduces complementary thermal cues that significantly improve robustness under extreme conditions. However, RGBT models are still restricted to closed-set categories and thus exhibit limited generalization. On the other hand, open-world detection (OWD) (Gu et al., 2021b; Zang et al., 2022; Zhong et al., 2022; Zang et al., 2022; Wu et al., 2024a) extends detectors beyond fixed taxonomies by leveraging vision–language alignment strategy such as CLIP (Radford et al., 2021) and GLIP (Li et al., 2022a), enabling recognition of novel categories in dynamic and evolving environments. Yet, OWD systems are typically trained on natural images and struggle with robustness when applied to challenging conditions. Therefore, current research reveals an inevitable trade-off: methods focusing on robustness often lack diversity, while methods emphasizing diversity compromise robustness.

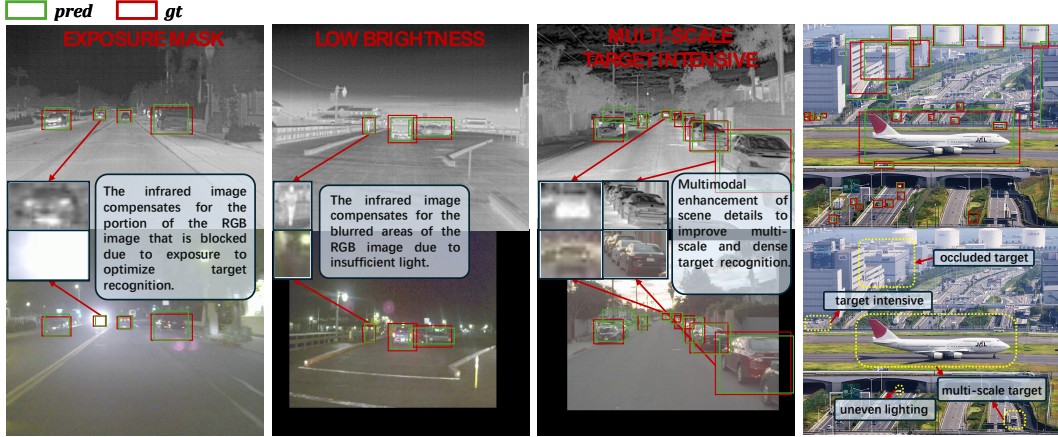

Figure 1: Visible-light imaging faces challenges like exposure mask, low brightness, and multi-scale target intensity, leading to information loss. Infrared imaging provides robust solutions, highlighting the necessity of multi-modal fusion.

*Can we develop a unified framework that simultaneously achieves both robustness under extreme conditions and open-vocabulary generalization, thereby breaking the current trade-off between diversity and robustness in object detection?*

In this paper, we propose a unified paradigm, **C3-OWD**, to simultaneously address both challenges through a curriculum cross-modal contrastive learning framework. The training is divided into two stages. In Stage 1, we enhance robustness by pretraining the detector on RGBT data such as FLIR dataset (Systems, 2018), enabling the model to resist severe environmental degradation. In Stage 2, we inject semantic priors from text via vision–language alignment, which improves generalization to unseen categories and strengthens the grounding of visual regions with textual descriptions.

A key difficulty in curriculum learning is catastrophic forgetting, where Stage 2 training may override the robustness learned in Stage 1. To alleviate this, we introduce an exponential moving average (EMA) mechanism (Karras et al., 2023; Lee et al., 2024a; Li et al., 2024). Our theoretical analysis demonstrates that the EMA mechanism preserves pre-stage performance with bounded parameter lag, ensuring function consistency between momentum and online branches and preventing catastrophic forgetting when adapting from small to large datasets.

Our main contributions are summarized as follows:

- We present C3-OWD, a unified cross-modal curriculum learning paradigm that integrates RGBT robustness with open-vocabulary generalization capabilities. Unlike traditional methods that suffer from modality bias, our approach dynamically balances multi-modal information through progressive learning, reducing the trade-off between robustness and diversity to achieve better adaptation to diverse environmental conditions.

- We provide rigorous theoretical foundations proving that our Exponential Moving Average (EMA) mechanism effectively prevents catastrophic forgetting with bounded parameter lag and guaranteed function consistency, establishing mathematical guarantees for stable knowledge retention in progressive multi-modal learning.

- Extensive experiments validate the effectiveness of C3-OWD, C3-OWD achieves $80.1$ $AP^{50}$ on FLIR, $48.6$ $AP^{50}_{Novel}$ on OV-COCO, and $35.7$ $mAP_r$ on OV-LVIS, achieving competitive results compared to prior state-of-the-art methods.

## 2 RELATED WORK

**Open-World Detection (OWD)** OWD focuses on detecting and learning unknown objects that are not annotated in the training set (Ma et al., 2023; Xi et al., 2024). During inference, detectors identify potential unknowns, which are then annotated and incrementally added as new categories.

Early methods, such as ORE (Joseph et al., 2021), enhanced Faster R-CNN with clustering and energy-based classifiers, while transformer-based approaches like OW-DETR (Gupta et al., 2022) adopted pseudo-labeling but often generated noisy labels. Extensions such as PROB (Zohar et al., 2023) introduced probabilistic objectness modeling, yet calibration issues remained. Recently, large visual models (e.g., SAM (Kirillov et al., 2023)) have further advanced OWOD through pseudo-labeling and knowledge distillation, though challenges in reliable unknown detection still persist.

**RGBT Object Detection** To address the limitations of relying solely on RGB images for object detection, some studies have introduced a thermal modality, leading to the development of RGBT object detection (Zhou et al., 2020; Sun et al., 2022a). Some methods use uncertainty or confidence metrics to balance RGB and thermal fusion (Kim et al., 2021; Li et al., 2023), while others adjust reliance on thermal inputs based on illumination levels (Guan et al., 2019; Li et al., 2019). Attention-based RGBT networks have also been proposed to further enhance modality integration (Shen et al., 2024a; Yuan & Wei, 2024). In this work, we employ RWKV as the backbone and design tailored training strategies to fully exploit the advantages of RGBT data.

**Receptance Weighted Key Value (RWKV)** RWKV (Peng et al., 2023) has emerged as a promising paradigm for sequential modeling, combining the efficiency of recurrent networks with the scalability of transformers (Li et al., 2025; Hou & Yu, 2024). By introducing time-shifted receptance gates and exponentially decaying key projections, it achieves linear-time complexity and efficient memory usage (Peng et al., 2025). These advantages have led to extensions in multimodal integration (Yang et al., 2025b; Fei et al., 2024). For example, PointRWKV (He et al., 2025) enhances geometric feature extraction in 3D point clouds, while Vision-RWKV (Duan et al., 2024b) improves high-resolution image understanding beyond traditional ViTs. In this work, we leverage RWKV to efficiently fuse two visual modalities and text, fully exploiting its potential in multimodal learning.

## 3 METHOD

### 3.1 OVERVIEW

Our goal is to enhance open-vocabulary detection under challenging conditions by introducing **C3-OWD**, a curriculum cross-modal contrastive learning framework. As illustrated in Fig. 2, our approach is built upon two-stage designed architecture. Stage 1 leverages RGBT (visible–thermal infrared) datasets to strengthen robustness. Stage 2 trains on COCO with Bi-Momentum Contrastive Alignment and Text-Modulated Deformabale Attention to achieve open-vocabulary detection with enhanced semantic grounding.

### 3.2 STAGE 1 - MULTI-MODAL ROBUSTNESS ENHANCEMENT

As shown in Fig 2 Stage 1, the primary objective of Stage 1 is to establish environmentally robust feature representations through complementary information from RGB and infrared modalities. Unlike traditional methods that employ simple feature concatenation or weighted fusion, we design an adaptive cross-modal interaction mechanism based on RWKV.

**Dual-Modal Feature Extraction:** Given paired RGB-infrared images $(I_{rgb}, I_{ir}) \in \mathbb{R}^{H \times W \times 3}$, we extract multi-scale features through a shared-weight CNN backbone:

$$\mathcal{F}_{rgb} = \{F_{rgb}^l\}_{l=2,3,4}, \quad \mathcal{F}_{ir} = \{F_{ir}^l\}_{l=2,3,4}. \tag{1}$$

where $F_*^l \in \mathbb{R}^{B \times C_l \times H_l \times W_l}$ denotes the feature map at layer $l$.

**VRWKV-based Adaptive Fusion:** Traditional RWKV employs unidirectional (forward) attention, which has limitations in RGB-Thermal cross-modal fusion: (1) **Asymmetric information flow**: Thermal features can compensate for information loss in low-light RGB regions, but unidirectional mechanisms prevent complementary information from flowing backward. (2) **Insufficient modal interaction**: As shown in Figure 1, when RGB images lose information due to exposure or occlusion, bidirectional flow is needed for effective modal compensation. To achieve efficient cross-modal interaction, we adopt VRWKV (Duan et al., 2024a) as the base architecture and extend it with a bidirectional attention mechanism to capture global dependencies. Our bidirectional mechanism introduces bidirectional position decay weights, enabling each token to consider both forward and

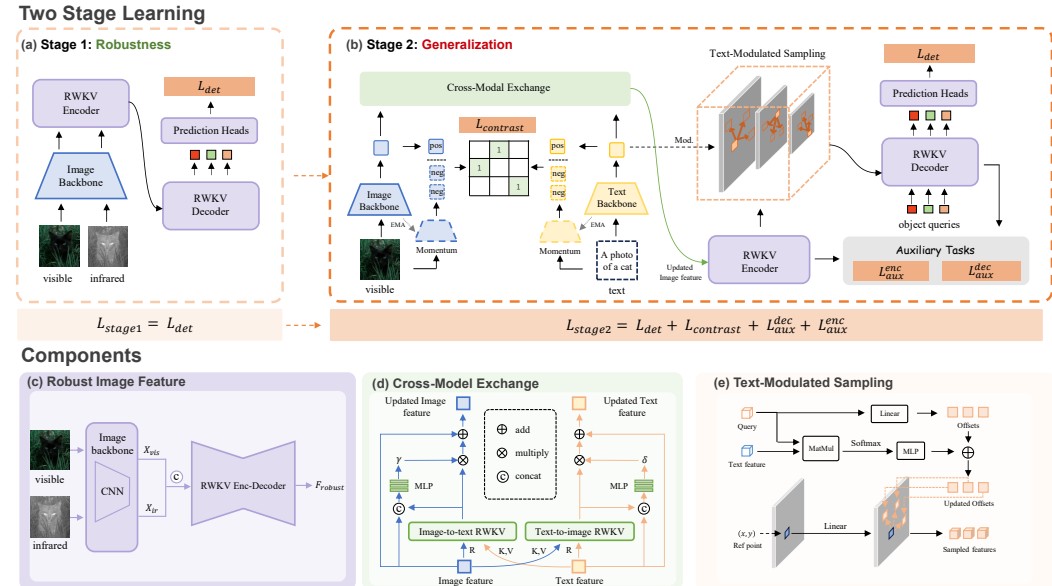

Figure 2: **Overview of C3-OWD**. (a) **Two-stage training**. Stage 1 leverages RGBT (visible–thermal infrared) datasets to strengthen robustness via RWKV. Stage 2 trains on COCO with Bi-Momentum Contrastive Alignment and Text-Modulated Deformabale Attention to achieve open-vocabulary detection with enhanced semantic grounding. (b) **Architecture components**. The Robust Image Feature Block performs efficient sequence modeling and multimodal feature fusion with linear-time complexity using the RWKV architecture. The cross-modal exchange mechanism enables bidirectional information flow and adaptive feature recalibration across modalities. The text-modulated sampling module steers deformable sampling locations with text features via attention, injecting stronger textual priors.

backward context, thus better capturing cross-modal complementarity.

For the $t$-th token, the attention output is computed as:

$$wkv_t = \text{Bi-WKV}(K,V)_t = \frac{\sum_{i=0,i\neq t}^{T-1} e^{-(|t-i|-1)/T\cdot w+k_i}v_i + e^{u+k_t}v_t}{\sum_{i=0,i\neq t}^{T-1} e^{-(|t-i|-1)/T\cdot w+k_i} + e^{u+k_t}}. \tag{2}$$

where $w$ is a learnable position decay parameter, $u$ is the current token importance weight, and $k_i$, $v_i$ are the key and value vectors respectively. This bidirectional mechanism allows each token to simultaneously consider both forward and backward contextual information.

RGB and Infrared features interact through the VRWKV-Block (Duan et al., 2024a) to generate modality-aware enhanced representations:

$$\mathcal{F}_{robust} = \text{VRWKV-Block}(\mathcal{F}_{rgb}, \mathcal{F}_{ir}). \tag{3}$$

where the VRWKV-block consists of an encoder-decoder architecture with cross-modal interaction mechanisms. The encoder processes multi-scale RGB-Thermal features, while the decoder refines the representations, enabling effective fusion of complementary information from both modalities.

During this stage, we employ standard detection losses including classification loss $\mathcal{L}_{cls}$ and regression loss $\mathcal{L}_{reg}$, directly supervised on paired RGB-infrared data. This pre-training strategy enables the model to learn robust feature representations across varying illumination conditions, establishing a strong foundation for subsequent open-world adaptation. The full details can be found in Algorithm 1.

### 3.3 STAGE 2 - VISION-LANGUAGE GENERALIZATION ALIGNMENT

As shown in Fig 2 Stage 2, in the second stage, we perform comprehensive vision-language alignment through three interconnected components: semantic enhancement fusion, text-modulated de-

formable attention, and bi-momentum contrastive learning. This stage enriches visual representations with semantic priors while maintaining efficient open-vocabulary detection capabilities.

### 3.3.1 Semantic Enhancement Fusion

We begin by enriching visual features with semantic priors through a hierarchical vision-language fusion module. Visual features $\mathbf{C} = \{C_2, C_3, C_4\}$ from the backbone (where $C_2 \in \mathbb{R}^{B \times 256 \times \frac{H}{8} \times \frac{W}{8}}$, $C_3 \in \mathbb{R}^{B \times 256 \times \frac{H}{16} \times \frac{W}{16}}$, $C_4 \in \mathbb{R}^{B \times 256 \times \frac{H}{32} \times \frac{W}{32}}$) are projected into a shared space via linear projection layers:

$$C_i' = \text{Linear}(C_i), \quad i \in \{2, 3, 4\} \tag{4}$$

Then, the scaled embeddings are flattened and concatenated to form the multi-scale visual sequence $\mathbb{V} = \text{Concat}(C_2', C_3', C_4') \in \mathbb{R}^{B \times L \times D}$, where $L$ is the total sequence length and $D$ is the embedding dimension. We then perform $l$ rounds of bidirectional cross-attention between the multi-scale visual features $\mathbb{V}$ and text embeddings $\mathbf{T_{clip}} \in \mathbb{R}^{B \times CLA \times D}$ (where $CLA$ is the number of classes), followed by cross-modal exchange:

$$\mathbb{V}_{out}, \mathbf{T}_{out} = \text{CrossModalExchange}(\mathbb{V}, \mathbf{T_{clip}}), \tag{5}$$

where the CrossModalExchange module performs bidirectional feature fusion using RWKV-based architecture. Specifically, the module computes cross-modal interactions through two parallel pathways:

$$\mathbf{V}_{fuse} = \text{RWKV}_{I \to T}(R = \mathbb{V}, K = \mathbf{T_{clip}}, V = \mathbf{T_{clip}}), \tag{6}$$

$$\mathbf{T}_{fuse} = \text{RWKV}_{T \to I}(R = \mathbf{T_{clip}}, K = \mathbb{V}, V = \mathbb{V}), \tag{7}$$

where $\text{RWKV}_{I \to T}$ denotes Image-to-Text RWKV interaction (updating visual features with text context) and $\text{RWKV}_{T \to I}$ denotes Text-to-Image RWKV interaction (updating text features with multi-scale visual context). Here, $R$ serves as the query-like input from the target modality while $K$ and $V$ are the key-value pairs from the source modality.

The attended features are then concatenated with the original features and processed through a two-layer MLP to generate gating coefficients:

$$\gamma = \text{MLP}([\mathbb{V} \cdot \mathbf{V}_{fuse}]), \quad \delta = \text{MLP}([\mathbf{T_{clip}} \cdot \mathbf{T}_{fuse}]), \tag{8}$$

where $\cdot$ denotes concatenation. The final outputs incorporate adaptive residual connections:

$$\mathbb{V}_{out} = \mathbb{V} + \gamma \otimes \mathbf{V}_{fuse}, \quad \mathbf{T}_{out} = \mathbf{T_{clip}} + \delta \otimes \mathbf{T}_{fuse}. \tag{9}$$

where $\otimes$ represents element-wise multiplication. This design enables adaptive cross-modal enhancement while effectively leveraging hierarchical visual representations.

### 3.3.2 Text-Modulated Sampling

In traditional methods, the sampling locations (where the model "looks") are predicted solely based on visual feature maps. However, for novel categories that the model has never seen, purely visual cues may be insufficient to locate the object of interest. To enhance spatial feature sampling with semantic guidance, we introduce a text-modulated sampling mechanism. Given query features $\mathbf{Q} \in \mathbb{R}^{B \times N \times D}$ and text features $\mathbf{T}_{text} \in \mathbb{R}^{B \times N \times D}$, the module first generates base sampling offsets:

$$\Delta_{base} = \text{Linear}(\mathbf{Q}), \tag{10}$$

where $\Delta_{base} \in \mathbb{R}^{B \times N \times 2K}$ represents $K$ sampling points with 2D offsets.

Simultaneously, text-guided modulation weights are computed through cross-modal attention:

$$\mathbf{A} = \text{Softmax}(\mathbf{Q} \cdot \mathbf{T}_{text}^T / \sqrt{D}), \mathbf{W}_{mod} = \text{MLP}(\mathbf{A}) \in \mathbb{R}^{B \times N \times 2K}, \tag{11}$$

where $\mathbf{A} \in \mathbb{R}^{B \times N \times C}$ represents attention scores between queries and text features, and the MLP maps the attention-weighted features from dimension $C$ to $2K$ to match the offset dimensions: $\text{MLP} : \mathbb{R}^{B \times N \times C} \to \mathbb{R}^{B \times N \times 2K}$.

The text-modulated offsets are obtained by combining base offsets with modulation:

$$\Delta_{updated} = \Delta_{base} \oplus \mathbf{W}_{mod}, \tag{12}$$

where $\oplus$ denotes element-wise addition. These updated offsets guide the sampling process:

$$\mathbf{F}_{sampled} = \text{Linear}(\text{Sample}(\mathbf{F}_{ref}, \mathbf{p}_{ref} + \Delta_{updated})). \tag{13}$$

where $\mathbf{F}_{ref}$ represents reference features, $\mathbf{p}_{ref} = (x, y)$ are reference points, and $\text{Sample}(\cdot)$ performs bilinear interpolation at the offset positions. This text-modulated sampling enables semantically-aware spatial attention, improving feature extraction at object boundaries and semantic regions.

### 3.3.3 BI-MOMENTUM CONTRASTIVE ALIGNMENT

To address the issue that negative samples are constrained by memory during training, we adopt MoCo (He et al., 2019) paradigm for region-text contrastive learning. Two momentum-updated feature queues are maintained:

$$\mathcal{Q}_{region} \in \mathbb{R}^{K \times D_{proj}}, \mathcal{Q}_{text} \in \mathbb{R}^{K \times D_{proj}}. \tag{14}$$

where $K$ is the queue size and $D_{proj}$ is the projection dimension.

Positive region proposals are selected via IoU threshold $\tau_{IoU}$:

$$\mathcal{P}_{pos} = \{p_i | \text{IoU}(p_i, g_j) \geq \tau_{IoU}, \exists g_j \in \mathcal{G}_{gt}\}. \tag{15}$$

Region features are encoded through RoI extraction and projection:

$$\mathbf{r}_q = f_\theta(\text{RoIExtractor}(\mathbf{F}_{out}, \mathcal{P}_{pos})), \mathbf{r}_k = f_{\theta_m}(\text{RoIExtractor}(\mathbf{F}_{out}, \mathcal{P}_{pos})). \tag{16}$$

where $f_\theta$ is a trainable projection network and $f_{\theta_m}$ is its momentum version updated via EMA:

$$\theta_m \leftarrow m \cdot \theta_m + (1 - m) \cdot \theta. \tag{17}$$

Multi-positive InfoNCE loss is used for image-to-text and text-to-image alignment:

$$\mathcal{L}_{i2t} = -\frac{1}{N_r} \sum_{i=1}^{N_r} \log \frac{\sum_{j \in \mathcal{P}_i^+} \exp(s_{ij}/\tau)}{\sum_{j=1}^{N_t+K} \exp(s_{ij}/\tau)}. \mathcal{L}_{t2i} = -\frac{1}{N_t} \sum_{j=1}^{N_t} \log \frac{\sum_{i \in \mathcal{P}_j^+} \exp(s_{ji}/\tau)}{\sum_{i=1}^{N_r+K} \exp(s_{ji}/\tau)}. \tag{18}$$

where $s_{ij} = \langle \mathbf{r}_{q,i}, \mathbf{t}_{k,j} \rangle \cdot \exp(\alpha)$ denotes the similarity between the $i$-th region feature and the $j$-th text feature, $\alpha$ is a learnable temperature parameter, $\tau$ is a fixed temperature hyperparameter, $N_r$ is the number of region features in the current batch, $N_t$ is the number of text features in the current batch, $K$ is the momentum queue size, $\mathcal{P}_i^+$ is the set of positive text indices for the $i$-th region, and $\mathcal{P}_j^+$ is the set of positive region indices for the $j$-th text. The total contrastive loss:

$$\mathcal{L}_{contrast} = \lambda_{i2t} \mathcal{L}_{i2t} + \lambda_{t2i} \mathcal{L}_{t2i}. \tag{19}$$

where $\lambda_{i2t}$ and $\lambda_{t2i}$ are loss weights. The full details can be found in Algorithm 2.

For more information regarding Stage 2, please refer to the appendix A.3

### 3.4 TWO-STAGE TRAINING STRATEGY

**Stage 1 - Multi-modal Robustness Training:** Using RGBT datasets, we enhance the model's robustness in extreme environments through visible-thermal infrared fusion. Only a single query head is used in this stage without any auxiliary heads. The loss function is defined as:

$$\mathcal{L}_{stage1} = \mathcal{L}_{det}(\mathbf{X}_{vis}, \mathbf{X}_{ir}). \tag{20}$$

where $\mathcal{L}_{det}$ is the detection loss, including classification loss $\mathcal{L}_{cls}$ and regression loss $\mathcal{L}_{reg}$.

**Stage 2 - Open-Vocabulary Training:** Training on the COCO dataset, we incorporate CLIP semantic features and momentum contrastive learning. Inspired by Co-DETR (Zong et al., 2023), we introduce multiple auxiliary heads to enhance the training efficiency of both encoder and decoder. The loss function is defined as:

$$\mathcal{L}_{stage2} = \mathcal{L}_{det} + \lambda_c \mathcal{L}_{contrast} + \lambda_{aux} \mathcal{L}_{aux}, \tag{21}$$

$$\mathcal{L}aux = \sum_{i=1}^{K} \mathcal{L}_i^{enc} + \lambda_1 \sum_{i=1}^{K} \sum_{l=1}^{L} \mathcal{L}_{i,l}^{dec}. \tag{22}$$

where $\mathcal{L}_{det}$ is the detection loss based on Hungarian matching including classification and regression losses, $\mathcal{L}_{contrast}$ is the contrastive loss for open-vocabulary learning using CLIP semantics, $\mathcal{L}_i^{enc}$ is the encoder auxiliary loss for the $i$-th head (e.g., ATSS or Faster R-CNN loss), $\mathcal{L}_{i,l}^{dec}$ is the decoder auxiliary loss for the $i$-th head at the $l$-th layer, $K$ is the number of auxiliary heads, $L$ is the number of decoder layers, $\lambda_c$ is the weight of $\mathcal{L}_{contrast}$, $\lambda_{aux}$ is the weight of $\mathcal{L}_{aux}$ and $\lambda_1$ is the weight for decoder auxiliary losses.

To mitigate catastrophic forgetting, as further detailed in Appendix A.8, which often arises in continual learning scenarios, we introduce Exponential Moving Average(EMA) (Karras et al., 2023; Lee et al., 2024a; Li et al., 2024). Intuitively, EMA maintains a momentum branch that smooths the parameter updates, keeping it close to the online branch while preserving stage-1 robustness. This helps ensure consistency of functions and loss stability across stages. A formal derivation and proof are provided in Appendix A.8.

The two-stage training strategy enables C3-OWD to first learn robust feature representations from multi-modal data, then leverage these representations to achieve superior open-vocabulary detection performance through semantic alignment with CLIP features and momentum contrastive learning.

## 4 EXPERIMENTS

### 4.1 DATASETS AND EVALUATION METRICS

The training process consists of two stages, utilizing the following datasets: the FLIR Dataset (Systems, 2018). The dataset provides thermal infrared images, a modality known for its high robustness to challenging conditions like extreme illumination (e.g., strong light, darkness) and noise pollution. They are used in the first training stage to enhance the detector's robustness and generalization capabilities by leveraging the invariant properties of the infrared spectrum. Subsequently, in the second training stage, we evaluate our approach on two standard open-vocabulary detection benchmarks modified from LVIS (Gupta et al., 2019) and COCO (Lin et al., 2014) respectively. LVIS (Gupta et al., 2019) contains 100K images with 1,203 classes. The classes are divided into three groups, namely frequent, common and rare, based on the number of training images. Following ViLD (Gu et al., 2021a), we treat 337 rare classes as novel classes and use only the frequent and common classes for training. The COCO dataset is a widely used benchmark for object detection, which consists of 80 classes. Following OVR-CNN (Zareian et al., 2021), we divide the classes in COCO into 48 base categories and 17 novel categories, while removing 15 categories without a synset in the WordNet hierarchy. The training set is the same as the full COCO but only images containing at least one base class are used. For FLIR (Systems, 2018) dataset, we report $AP^{50}$—the mean Average Precision (mAP) at an IoU threshold of 0.5 for base categories—as the primary metric. For COCO (Lin et al., 2014), we report $AP_{novel}^{50}$—the mean Average Precision (mAP) at an IoU threshold of 0.5 for novel categories—as the primary metric. Additionally, we provide performance on base categories ($AP_{base}^{50}$) and overall performance across all categories ($AP^{50}$). For LVIS (Gupta et al., 2019), we report $AP_r$, $AP_c$, and $AP_f$—denoting mAP on rare, common, and frequent categories, respectively—along with the overall $AP$, all computed using standard box-based mAP.

### 4.2 IMPLEMENTATION DETAILS

Our model is based on a Deformable-DETR (Zhu et al., 2020) architecture, initialized from a pretrained ResNet-50 backbone. The first-stage training utilizes the FLIR dataset (Systems, 2018), while the second-stage is performed on the COCO (Lin et al., 2014) and LVIS Gupta et al. (2019) dataset. Both stage-1 and stage-2 trainings use 36 epochs. We employ the AdamW optimizer with an initial learning rate of $5 \times 10^{-5}$ and weight decay of $1 \times 10^{-4}$, applying a layer-wise learning rate decay with a multiplier of 0.1 for the backbone. The learning rate is reduced by a factor of 10 at the $12^{th}$ and $24^{th}$ epoch during the 36-epoch training schedule.Following the MoCo (He et al., 2019), We set $K = 65536$ and $\tau_{IoU} = 0.3$. All input images are resized to $640 \times 640$ pixels with a batch size of 4 per GPU, and we apply gradient clipping with a maximum norm of 0.1. Following Co-DTER Zong et al. (2023), the model incorporates multiple detection heads including

Table 1: Comparison on FLIR dataset.

| Method | Backbone | AP$^{50}$ (FLIR) |
|---|---|---|
| MMTOD-CG (Devaguptapu et al., 2019) | RN50 | 61.4 |
| MMTOD-UNIT (Devaguptapu et al., 2019) | RN101 | 61.5 |
| CMPD (Li et al., 2022b) | RN50 | 69.4 |
| CFR (Zhang et al., 2020) | VGG-16 | 72.4 |
| GAFF (Zhang et al., 2021) | RN50 | 72.9 |
| BU-ATT (Kieu et al., 2021) | RN50 | 73.1 |
| BU-LTT (Kieu et al., 2021) | RN50 | 73.2 |
| UA_CMDet (Sun et al., 2022b) | RN50 | 78.6 |
| CFT (Fang et al., 2021) | RN50 | 78.7 |
| CSAA (Cao et al., 2023) | RN50 | 79.2 |
| ICAFusion (Shen et al., 2024b) | RN50 | 79.2 |
| CrossFormer (Lee et al., 2024b) | Swin-B | 79.3 |
| MFPT (Zhu et al., 2023) | RN101 | 80.0 |
| RSDet (Zhao et al., 2024b) | RN50 | 81.1 |
| MiPa (Medeiros et al., 2024) | ViT-B/16 | 81.3 |
| MMFN (Yang et al., 2025a) | RN50 | **81.8** |
| **C3-OWD (Ours)** | RN50 | 80.1 |

RPN, query-based, and ROI heads, with a total of 6 encoder layers and 8 decoder layers, the loss weighting factor $\lambda_2$ set to 2.0. We set $\lambda_c = 0.01$, $\lambda_{aux} = 0.1$, and $\lambda_1 = 1$. The temperature $\tau$ is set to 0.07. We train and evaluate our model on $8 \times$NVIDIA A100 GPUs.

## 4.3 MAIN RESULT

As shown in Table 1, 2. Although C3-OWD does not achieve the absolute state-of-the-art on the FLIR dataset (slightly behind MMFN's 81.8), it significantly outperforms all existing methods on open-vocabulary detection tasks (OV-COCO and OV-LVIS), notably achieving a remarkable AP$^{50}_{Novel}$ of 48.6 on OV-COCO—surpassing the previous best result of 44.3 by CLIPSelf. More importantly, C3-OWD is the method that demonstrates leading performance across both multimodal robustness and open-vocabulary detection tasks, highlighting its superior generality and cross-task adaptability. This indicates that our proposed curriculum cross-modal contrastive learning framework not only enhances robustness in extreme environments but also effectively transfers to open-vocabulary scenarios, enabling broader applicability.

Table 2: **Comparison with state-of-the-art methods on OV-COCO and OV-LVIS benchmarks.** We report AP$^{50}_{Novel}$ for OV-COCO and mAP$_r$ (rare classes) for OV-LVIS. "-" indicates the result is not reported in the original paper for the specific backbone. **Bold** indicates the best performance.

| Method | Backbone | OV-COCO (AP$^{50}_{Novel}$) | OV-LVIS (mAP$_r$) |
|---|---|---|---|
| *ResNet-50 Based Methods* | | | |
| ViLD (Gu et al., 2021b) | RN50 | 27.6 | 16.3 |
| Detic (Zhou et al., 2022) | RN50 | 27.8 | 24.9 |
| OV-DETR (Zang et al., 2022) | RN50 | 29.4 | 17.4 |
| ProxyDet (Jeong et al., 2024) | RN50 | 30.4 | - |
| RegionCLIP (Zhong et al., 2022) | RN50 | 31.4 | 17.1 |
| RTGen (Chen et al., 2024) | RN50 | 33.6 | 21.4 |
| BARON-KD (Wu et al., 2023a) | RN50 | 34.0 | 22.6 |
| CLIM (Wu et al., 2024b) | RN50 | 36.9 | 23.9 |
| SAS-Det (Zhao et al., 2024a) | RN50 | 37.4 | 20.9 |
| *Larger ResNet Variants* | | | |
| RegionCLIP (Zhong et al., 2022) | RN50x4 | 39.3 | 22.0 |
| CORA (Wu et al., 2023b) | RN50x4 | 41.7 | 28.1 |
| SAS-Det (Zhao et al., 2024a) | RN50x4 | 43.9 | 29.1 |
| CLIM (Wu et al., 2024b) | RN50x64 | - | 32.3 |
| F-VLM (Kuo et al., 2023) | RN50x64 | - | 32.8 |
| *Transformer Based Methods (ViT / Swin)* | | | |
| PromptDet (Song & Bang, 2023) | ViT-B/16 | 30.6 | - |
| RTGen (Chen et al., 2024) | Swin-B | - | 30.2 |
| BIND (?) | ViT-L/16 | 41.5 | 32.5 |
| Detic (Zhou et al., 2022) | Swin-B | - | 33.8 |
| CFM-ViT (Kim et al., 2023a) | ViT-L/16 | 34.1 | 33.9 |
| RO-ViT (Kim et al., 2023b) | ViT-H/16 | 33.0 | 34.1 |
| CLIPSelf (Wu et al., 2024a) | ViT-L/14 | 44.3 | 34.9 |
| ProxyDet (Jeong et al., 2024) | Swin-B | - | 36.7 |
| CoDet (Ma et al., 2024) | ViT-L/14 | 44.7 | **37.0** |
| **C3-OWD (Ours)** | **RN50** | **48.6** | 35.7 |

Table 3: Ablation study of C3-OWD components on OV-COCO and OV-LVIS datasets.

| No. | Ablated Component | OV-COCO ($AP_{novel}^{50}$) | OV-LVIS ($AP_r$) |
|---|---|---|---|
| 0 | C3-OWD (Full Model) | **48.6** | **35.7** |
| 1 | w/o encoder fusion | 42.1 | 31.2 |
| 2 | static query selection | 44.1 | 32.3 |
| 3 | w/o text cross-attention | 40.9 | 29.8 |
| 4 | w/o MoCo | 39.7 | 27.5 |
| 5 | w/o Bi-MoCo | 46.1 | 33.2 |

Table 4: Ablation study on Stage-1 modality and Stage-2 initialization.

| Config. | Stage-1 Modality | Stage-2 Initialization | FLIR ($AP^{50}$) | OV-COCO ($AP_{novel}^{50}$) | OV-LVIS ($AP_r$) |
|---|---|---|---|---|---|
| Full Model | RGB + IR | From Stage-1 | **80.1** | **48.6** | **35.7** |
| A | RGB only | From Stage-1 | 75.2 | 43.5 | 32.1 |
| B | IR only | From Stage-1 | 76.8 | 44.1 | 32.8 |
| C | RGB + IR | Random | 80.0 | 41.2 | 30.5 |
| D | RGB only | Random | 75.1 | 39.8 | 29.1 |
| E | IR only | Random | 76.7 | 40.3 | 29.7 |

## 4.4 ABLATION STUDY

Table 3 presents the ablation results of **key components** in C3-OWD. The full model (No. 0) achieves the best performance on both OV-COCO and OV-LVIS datasets. Removing the encoder fusion module (No. 1) causes significant performance drops (6.5 in $AP_{novel}^{50}$ and 4.5 in $AP_r$), demonstrating the crucial role of hierarchical feature fusion for open-vocabulary detection. Using static query selection (No. 2) instead of deformable attention also leads to considerable performance degradation, validating the effectiveness of dynamic query optimization. The absence of text cross-attention (No. 3) results in the most substantial performance decline, indicating that vision-language interaction is core to cross-modal alignment. Removing MoCo contrastive learning (No. 4) hurts performance the most, highlighting the critical importance of momentum contrast in addressing positive-negative sample imbalance. Finally, using single-queue instead of dual-queue MoCo (No. 5) also reduces performance, proving that the region-text dual-queue design effectively prevents inter-modal interference. All ablation results consistently show that each component in C3-OWD contributes importantly to the final performance.

As shown in Table 4, we conduct a thorough ablation study to validate the necessity of the two-stage curriculum design in C3-OWD. The results clearly demonstrate that both the multimodal training in Stage-1 and the weight inheritance in Stage-2 are critical for achieving optimal performance. Using only a single modality (RGB or IR) in Stage-1 leads to a noticeable drop in robustness, as evidenced by the decreased FLIR $AP^{50}$ (Configs A and B vs. Full). This performance degradation propagates to the open-vocabulary detection tasks in Stage-2, underscoring the importance of cross-modal complementary learning for acquiring generalized representations. Initializing Stage-2 with random weights instead of the pre-trained weights from Stage-1 results in a substantial performance collapse on OV-COCO and OV-LVIS (Config C vs. Full), even when the full RGB-IR modality was used previously. This indicates that the robust features learned in Stage-1 are a crucial foundation for the subsequent semantic alignment and contrastive learning. The worst performance is observed when both components are ablated (Configs D and E), reinforcing that the proposed curriculum learning pipeline is holistic and both stages are indispensable.

As shown in Table 5, we compared Inference FPS on a single A100 GPU with input size $640 \times 640$ and the batch size is 8. Thanks to the RWKV-based fusion and efficient attention, our method achieves **62.4 FPS**, meeting real-time requirements for autonomous systems. Although the incorporation of MoCo contrastive learning increases the model parameter count, leading to a 17% decrease in inference speed, the resulting improvement in accuracy and enhanced multimodal fusion performance make this computational cost acceptable. We chose $\tau_{IoU} = 0.3$ specifically for the Open-World setting. In open-vocabulary detection, the Region Proposal Network (RPN) often assigns lower confidence scores to novel/unseen categories compared to base categories. A standard threshold (e.g., 0.5) tends to filter out valid proposals for novel objects, hurting Recall.

Table 5: Inference Speed and Memory Comparison (COCO)

| Method | Backbone | Complexity | Params (M) | FPS |
|---|---|---|---|---|
| RegionCLIP | ResNet-50 | $O(N^2)$ | 216 | 68.47 |
| OV-DETR | ResNet-50 | $O(N^2)$ | 165 | 71.59 |
| RegionCLIP | ResNet-50x4 | $O(N^2)$ | 403 | 52.21 |
| CLIM | ResNet-50x4 | $O(N^2)$ | 204 | 69.8 |
| Detic | Swin-B | $O(N^2)$ | 670 | 46 |
| CLIPSelf | ViT/L-14 | $O(N^2)$ | 480 | 46.2 |
| **C3-OWD (w/o MoCo)** | ResNet-50 | $O(N)$ | **124.3** | **73.1** |
| **C3-OWD (Ours)** | ResNet-50 | $O(N)$ | 245.6 | 62.4 |

As shown in Table 6, decreasing IoU from 0.5 to 0.3 significantly improves $AP_{Novel}$ (+3.4) and

Table 6: Ablation of IoU Threshold ($\tau_{IoU}$) on OV-COCO

| $\tau_{IoU}$ | $AP^{50}_{Novel}$ | $AP^{50}_{Base}$ | Recall (Novel) |
|---|---|---|---|
| 0.5 | 45.2 | **56.8** | 62.4% |
| 0.4 | 47.1 | 56.5 | 68.9% |
| **0.3 (Ours)** | **48.6** | 56.1 | **74.3%** |
| 0.2 | 48.2 | 54.9 | 71.1% |

Recall (+11.9%) with only a negligible drop in Base AP. Setting it too low (0.2) introduces excessive noise. Thus, 0.3 provides the optimal trade-off.

Table 7: Impact of EMA on Preserving Stage-1 Robustness

| Setting | **FLIR $AP^{50}$ (Robustness)** | OV-COCO $AP_{Novel}$ |
|---|---|---|
| **C3-OWD (Full with EMA)** | **80.1** | **48.6** |
| w/o EMA (Direct Finetuning) | 59.5 (-20.6) | 47.8 (-0.8) |
| w/o Momentum Contrast | 72.3 (-7.8) | 39.7 (-8.9) |

**Role in Robustness (FLIR):** As shown in Table 7, removing the EMA mechanism (i.e., allowing the model to update freely without a momentum teacher) causes the FLIR performance to drop significantly from 80.1 to 59.5. This empirically validates our Theorem 1, showing that EMA is essential to enforce function consistency and retain the robust RGBT features learned in Stage 1. **Role in Generalization (COCO):** While the Open-Vocabulary performance on COCO remains relatively high without EMA (47.8), the model sacrifices its unique advantage—robustness. Therefore, the ablation should be seen as proof that EMA is the "stabilizer" that allows C3-OWD to excel in both tasks simultaneously, rather than trading one for the other.

## 5 CONCLUSION

In this paper, we introduce an innovative curriculum cross-modal contrastive learning framework for open-world detection, C3-OWD. The proposed two-stage training paradigm addresses the fundamental challenge of achieving both environmental robustness and open-world generalization. Our EMA mechanism provides theoretical guarantees for knowledge preservation across training stages, with rigorously proven bounded parameter lag and function consistency. The integration of RGBT multimodal fusion with RWKV-based architecture enables effective capture of complementary information from visible and infrared modalities, while the vision-language alignment stage extends detection capabilities to novel categories without sacrificing the learned robustness. Extensive experiments validate the effectiveness of C3-OWD across OV-COCO,OV-LVIS and FLIR. C3-OWD achieves competitive results compared to prior state-of-the-art methods.

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

# A APPENDIX

## A.1 USE OF LLM

We use LLM to aid or polish writing.

## A.2 ETHICS STATEMENT

This work adheres to the ICLR Code of Ethics. Our study does not involve human subjects, personal or sensitive data. All datasets used in this paper (e.g., COCO, LVIS, and public RGBT benchmarks) are publicly available and widely adopted in the research community, and we strictly follow their licenses and intended usage. The proposed C3-OWD framework is designed for academic exploration of open-vocabulary detection under challenging conditions. Potential misuse of the model in safety-critical or surveillance scenarios is outside the scope of this research, and we strongly encourage responsible and ethical use in line with research integrity principles.

## A.3 METHODOLOGICAL DETAILS

In this section, we provide a more intuitive and detailed walkthrough of the two key components in Stage 2: *Semantic Enhancement Fusion* and *Text-Modulated Sampling*. These explanations complement the formal definitions in the main text (Section 3.3) by focusing on the design rationale.

### A.3.1 DETAILS OF SEMANTIC ENHANCEMENT FUSION

In standard open-vocabulary detectors, visual features extracted from the backbone are often spatially rich but semantically "silent"—they describe textures and edges but lack knowledge of the categories (e.g., "dog", "car") they represent. This creates a misalignment when these features meet the text embeddings later in the pipeline. The **Semantic Enhancement Fusion** module aims to bridge this gap *early* in the encoding stage. It essentially "paints" the visual feature maps with semantic concepts derived from the text embeddings, ensuring that the visual representation carries semantic priors before entering the detection head.

- **Inputs:** The module receives two streams:
    1. The flattened visual sequence $\mathbb{V}$, derived from multi-scale feature maps $(C_2, C_3, C_4)$. In the context of RWKV, this sequence represents the visual context.
    2. The text embeddings $T_{clip}$, which act as a dictionary of potential object categories (e.g., the embeddings for "person", "bicycle", etc.).

- **The Interaction Mechanism (Cross-Modal Exchange):** Unlike standard cross-attention which has quadratic complexity $O(N^2)$, we leverage the linear complexity of the RWKV architecture to perform efficient bidirectional exchange. As defined in Eq. 6 and 7, this process can be understood as two simultaneous updates:
    - **Image-to-Text** ($I \rightarrow T$)**:** The text features query the image to see which categories are actually present in the scene. This highlights relevant class embeddings.
    - **Text-to-Image** ($T \rightarrow I$)**:** The visual pixels query the text dictionary. If a visual region resembles a "cat", it pulls information from the "cat" text embedding, enhancing its own representation.

Directly injecting text features into visual maps can sometimes introduce noise (e.g., text concepts not present in the image). To mitigate this, we employ the adaptive gating mechanism described in Eq. 8 and 9. The gating coefficients $\gamma$ and $\delta$ act as "valves." If the interaction suggests a strong correlation between a visual region and a text concept, the valve opens ($\gamma$ is high), allowing the semantic information to flow into the visual feature $\mathbb{V}_{out}$. Conversely, for background regions irrelevant to any text description, the valve closes, preserving the original visual information. This ensures that $\mathbb{V}_{out}$ is a semantically enriched version of the original backbone features, primed for the subsequent detection tasks.

### A.3.2 Details of Text-Modulated Sampling

Our architecture builds upon Deformable-DETR. In the original Deformable-DETR, the model predicts "sampling offsets" to decide where to look on the feature map. Typically, these sampling locations are predicted solely based on the query's current content (visual appearance). However, in an Open-World setting, the model may encounter a novel category it has never seen before. Relying only on visual cues might fail to focus on the discriminative parts of this unknown object. We introduce **Text-Modulated Sampling** to use the text description as a "flashlight," guiding the model's gaze toward regions that match the semantic description. **Mechanism Explanation.** The process modifies how the *sampling points* ($p_{ref} + \Delta$) are generated in the Deformable Attention layer.

- **The Query** ($Q$)**:** In DETR-like models, $Q$ represents a "detective" searching for an object.
- **Base Offsets** ($\Delta_{base}$)**:** Defined in Eq. 10, this is the standard Deformable-DETR behavior. The detective looks at locations based on visual patterns (e.g., looking for corners or blobs).
- **Text Guidance** ($W_{mod}$)**:** Defined in Eq. 11, we compute the similarity between the Query $Q$ and the Text Embeddings $T_{text}$. If a query has high affinity with the text embedding for "giraffe", the network generates a modulation weight $W_{mod}$ specific to "giraffe-like" features.

The core innovation lies in Eq. 12: $\Delta_{updated} = \Delta_{base} \oplus W_{mod}$. Intuitively, $\Delta_{base}$ provides the initial guess of the object's shape. The term $W_{mod}$ acts as a correction vector driven by semantics. For instance, if the text query is "tall animal with long neck," the $W_{mod}$ shifts the sampling points vertically to cover the neck region, even if the visual features alone were ambiguous. Finally, the function $Sample(\cdot)$ in Eq. 13 performs bilinear interpolation at these *shifted* coordinates. By physically moving the sampling locations, we ensure that the features extracted for the subsequent prediction heads ($F_{sampled}$) contain information that is not only visually salient but also semantically aligned with the target text description.

### A.4 Visualization

We conducted a comparative analysis with traditional methods, as illustrated in Fig 3. Conventional RGBT detectors (e.g., ICAFusion (Shen et al., 2024b), MFPT (Zhu et al., 2023), etc.) are constrained by their closed-set nature, limiting their generalization in open-world scenarios with numerous object categories. On the other hand, traditional open-vocabulary object detection (OVOD) methods (e.g., ViLD (Gu et al., 2021b), CORA (Wu et al., 2023b), etc.) rely solely on the RGB modality and lack complementary infrared information, leading to performance degradation under challenging conditions such as occlusion and shadow coverage. In contrast, our proposed detector effectively integrates multi-modal inputs with open-vocabulary recognition, demonstrating superior generalization across diverse complex environments. Moreover, comparisons between RGB and thermal infrared modalities reveal that while all models perform well under infrared imaging, conventional OVOD detectors exhibit significant performance drops in RGB under low-light or overexposed conditions, further highlighting the enhanced adaptability and generality of our model.

### A.5 Reproducibility Statement

We make every effort to ensure the reproducibility of our results. Full training details, including model architectures, hyperparameters, and optimization schedules, are provided in the main paper and appendix. The experimental settings cover both Stage 1 (RGBT-based robustness enhancement) and Stage 2 (open-vocabulary detection with CLIP and contrastive learning), with clear descriptions of dataset preprocessing and evaluation protocols. Our implementation is based on PyTorch and standard detection frameworks. To facilitate replication, we will release the source code, configuration files, and pre-trained models upon publication. All reported results can be reproduced using the provided settings and supplementary material.

### A.6 Ablation

To demonstrate that C3-OWD generalizes beyond the specific modality (RGBT) used in Stage 1, we evaluated our model on other datasets: DAWN (Adverse Weather), RTTS (Haze/Fog), VisDrone (Aerial/Small Objects).

Table 8: Performance comparison on additional adverse environment datasets ($mAP^{50}$).(**\*** denotes the metric obtained from our reproduced results.)

| Method | Adverse Weather | | Aerial |
| | DAWN | RTTS | VisDrone |
| --- | --- | --- | --- |
| CMC(YOLOv5n) (Raza et al., 2024) | **72.6** | 66.2* | 17.8* |
| YOLOv8-STE (Jing et al., 2024) | 68.26 | **74.51** | 20.1* |
| Dome-DETR (Hu et al., 2025) | 65.32* | 68.3* | **33.52** |
| **C3-OWD (Ours)** | **71.3** | **73.31** | **27.83** |
| *Changes vs SOTA* | *-1.3* | *-1.2* | *-5.69* |

Since these datasets are predominantly RGB-only, these experiments validate the transferability of the robust feature representations learned during our Stage 1. As shown in Table 8, C3-OWD consistently outperforms strong baselines:

- **Robustness Transfer:** On DAWN (Kenk & Hassaballah, 2020) and RTTS (Li et al., 2018), our method achieves **71.3** and **73.31** mAP, respectively, showcasing strong robustness transfer capability under adverse weather conditions.

- **Domain Generalization:** On VisDrone (Zhu et al., 2021) which represents severe domain shifts, our method maintains competitive performance, demonstrating that the RWKV-based fusion and CLIP alignment provide a strong generalizable prior.

### A.7 LIMITATIONS

Despite the promising results, our method suffers from several limitations that warrant further investigation. **Computational Overhead.** The hierarchical cross-modal fusion mechanism, while effective, introduces additional computational complexity due to the bidirectional RWKV blocks and iterative attention operations. This may hinder real-time deployment on resource-constrained devices such as embedded systems or drones. **Dependency on Multi-modal Data.** The first stage relies heavily on aligned RGB-Thermal (RGBT) data for robustness enhancement. Such paired data is scarce and expensive to collect and annotate, limiting the scalability of our approach in domains where thermal imagery is unavailable. **Generalization to Unseen Modalities.** Although our method improves open-vocabulary detection, its performance on entirely unseen sensor modalities (e.g., LiDAR, radar) or under extreme domain shifts remains unverified. **Training Complexity.** The two-stage training strategy requires carefully designed curricula and hyperparameter tuning, which may increase the risk of suboptimal convergence and complicate reproduction. **Semantic Granularity.** While CLIP provides rich semantic prior, its knowledge is constrained by pre-trained concepts. Our method may still struggle with highly fine-grained or domain-specific categories absent in CLIP's training distribution. We believe these limitations point to meaningful directions for future work, including efficient fusion design, self-supervised adaptation, and generalized multi-modal pretraining.

### A.8 THEOREM AND PROOF

**Theorem 1** (EMA preserves pre-stage performance for the momentum branch). *Let $\{\theta_t\}_{t \geq 0} \subset \mathbb{R}^p$ be the online-branch parameters during Stage-2 training, and let the momentum (EMA) branch be updated by $\theta_{m,t} \leftarrow m\,\theta_{m,t-1} + (1-m)\,\theta_t$ and $\theta_{m,0} = \theta_0$ with $m \in (0,1]$. Denote by $f_\theta$ (online) and $f_{\theta_m}$ (momentum) the projection networks used to produce region/text embeddings, and let the multi-positive InfoNCE loss be $\mathcal{L}_{\mathrm{contrast}}(\theta)$ (with similarities $s_{ij} = \langle r_i(\theta), t_j(\theta) \rangle$ and temperature $\tau > 0$). Here, we assume:*

- *(A1) (Bounded per-step motion) $\|\theta_t - \theta_{t-1}\| \leq \delta_t$ for all $t \geq 1$, and write $\Delta_t := \max_{1 \leq j \leq t} \delta_j$.*

- *(A2) (Parameter-to-feature Lipschitz) For any RoI/text input, there exist $K_r, K_t > 0$ such that $\|r_i(\theta) - r_i(\theta')\| \leq K_r \|\theta - \theta'\|$ and $\|t_j(\theta) - t_j(\theta')\| \leq K_t \|\theta - \theta'\|$ for all $\theta, \theta'$ in a convex set containing $\{\theta_s, \theta_{m,s}\}_{s \leq t}$.*

*(A3) (Logit/Loss Lipschitz) With cosine (or normalized) similarity $s_{ij} = \langle r_i, t_j \rangle$ and temperature $\tau$, the InfoNCE per-example loss is $\rho$-Lipschitz w.r.t. the logit vector, with $\rho \leq 1/\tau$.*

*Then, for every $t \geq 1$, the momentum–online parameter lag satisfies*

$$\|\theta_t - \theta_{m,t}\| \leq \frac{1-m}{m} \Delta_t, \tag{23}$$

*and the following bounds hold:*

$$\text{(Function-consistency)} \quad \mathbb{E}\big[\|f_{\theta_t}(x) - f_{\theta_{m,t}}(x)\|\big] \leq (K_r + K_t) \frac{1-m}{m} \Delta_t, \tag{24}$$

$$\text{(Loss preservation)} \quad \big|\mathcal{L}_{\text{contrast}}(\theta_{m,t}) - \mathcal{L}_{\text{contrast}}(\theta_t)\big| \leq \rho (K_r + K_t) \frac{1-m}{m} \Delta_t. \tag{25}$$

*In particular, deploying the EMA/momentum parameters $\theta_{m,t}$ at any time $t$ guarantees an $\varepsilon$-tolerance on the reference distribution whenever $\frac{1-m}{m} \Delta_t \leq \frac{\varepsilon}{\rho (K_r + K_t)}$.*

*Proof.* **(EMA lag).** From the update $\theta_{m,t} = m\theta_{m,t-1} + (1-m)\theta_t$, subtract from $\theta_t$: $\theta_t - \theta_{m,t} = (1-m)\big(\theta_t - \theta_{m,t-1}\big) = (1-m)\big[(\theta_t - \theta_{t-1}) + (\theta_{t-1} - \theta_{m,t-1})\big]$. Taking norms and using the triangle inequality gives $x_t := \|\theta_t - \theta_{m,t}\| \leq (1-m) u_t + (1-m) x_{t-1}$ and $u_t := \|\theta_t - \theta_{t-1}\| \leq \Delta_t$. Unrolling with $x_0 = 0$ yields $x_t \leq (1-m) \sum_{k=0}^{t-1} (1-m)^k u_{t-k} \leq (1-m) \Delta_t \sum_{k=0}^{\infty} (1-m)^k = \frac{1-m}{m} \Delta_t$, which is Eq. (23).

**(Function-consistency).** By (A2), for any input (RoI/text) $x$, $\|f_{\theta_t}(x) - f_{\theta_{m,t}}(x)\| \leq (K_r + K_t) \|\theta_t - \theta_{m,t}\|$. Taking expectation over the reference distribution and using Eq. (23) gives Eq. (24).

**(Loss preservation).** With normalized features, the similarity logits enter InfoNCE as $\{s_{ij}/\tau\}$. By (A3), the per-example loss is $\rho$-Lipschitz in logits, hence the loss difference between $\theta_t$ and $\theta_{m,t}$ is bounded by $\rho \cdot \mathbb{E}\big[\|\text{logits}(\theta_t) - \text{logits}(\theta_{m,t})\|\big]$. Since each logit is an inner product of two unit vectors, its perturbation is bounded by the sum of the feature perturbations; with (A2) and Eq. (23) this expectation is at most $(K_r + K_t) \frac{1-m}{m} \Delta_t$, proving Eq. (25). $\square$

**Catastrophic forgetting** manifests as a sharp decline in robustness metrics (e.g., FLIR AP) after the model is trained on the second stage (COCO/LVIS). This is caused by the **"plasticity-stability" dilemma**: the optimization for the open-vocabulary objective ($\mathcal{L}_{\text{stage2}}$) shifts the parameters away from the RGBT-robust manifold learned via ($\mathcal{L}_{\text{stage1}}$). The **EMA mechanism** maintains a "slow" momentum branch ($\theta_m$) that acts as a memory of the robust features. We provide a theoretical justification by back-deducing the bound on the **KL divergence** based on the paper's formulas. Let $P_{\theta_t}$ be the distribution of the online model at step $t$ and $P_{\theta_{m,t}}$ be the distribution of the momentum model. The extent of "forgetting" can be measured by the divergence of the current model from the robust momentum trajectory, i.e., $D_{\text{KL}}(P_{\theta_{m,t}} || P_{\theta_t})$.

**Parameter Lag Bound (From Eq. 23 in Theorem 1, Appendix A.8)** The parameter lag between the online parameters and the EMA parameters is bounded as:

$$\|\theta_t - \theta_{m,t}\| \leq \frac{1-m}{m} \cdot \Delta_t$$

where $m$ is the EMA momentum coefficient and $\Delta_t$ denotes the maximum update step (in norm) at iteration $t$.

**KL Divergence Approximation (Second-Order Taylor Expansion)** Assuming the loss function (negative log-likelihood) is twice differentiable and locally smooth with Hessian ($H$), the KL divergence between the corresponding predictive distributions can be approximated by the second-order Taylor expansion:

$$D_{\text{KL}}(P_{\theta_{m,t}} || P_{\theta_t}) \approx \frac{1}{2} (\theta_t - \theta_{m,t})^T H (\theta_t - \theta_{m,t})$$

**Bounding the KL Divergence** Let $L$ be the Lipschitz constant of the gradient (boundedness of $\|H\|_2$). Substituting the bound from Eq. 23:

$$D_{\text{KL}}(P_{\theta_{m,t}} || P_{\theta_t}) \leq \frac{1}{2} L \|\theta_t - \theta_{m,t}\|^2 \leq \frac{1}{2} L \left( \frac{1-m}{m} \cdot \Delta_t \right)^2$$

This derived inequality proves that the **semantic drift (forgetting)** between the EMA model and the online model is **quadratically bounded** by the EMA coefficient. By setting $m \approx 1$ (like 0.999), we strictly limit the upper bound of the KL divergence, thereby theoretically guaranteeing the preservation of Stage 1 robustness.

### A.9 FAILURE CASES

Despite strong overall performance, C3-OWD occasionally fails in challenging scenarios. As shown in Fig 4.(a), Dense objects without internal heat sources, such as hats, tables, and chairs, have significantly affected the detection accuracy, and our current model struggles to reliably detect them, highlighting the sensitivity of IR-based detection to environmental heat sources. In Fig 4.(b), a highly occluded vehicle in a dense traffic scene is missed due to insufficient visible cues, suggesting limitations in cross-modal inference under severe occlusion. Fig 4.(c) shows a fine-grained recognition error where a "husky" is misclassified as "dog", indicating that CLIP's semantic prior may not capture subtle inter-class distinctions.

### A.10 FUTURE WORK

Based on the failure cases and limitations identified in Sections A.7 and A.9, we outline several promising directions for future research. **Self-Supervised Modality Alignment.** Reducing dependency on paired RGBT data requires methods that can learn robust representations from unaligned or weakly-aligned multi-modal streams. Contrastive learning frameworks (Radford et al., 2021) that align modalities in a shared embedding space without strict pixel-level correspondence offer a viable path forward. **Occlusion-Robust Representation Learning.** The failure cases under severe occlusion suggest the need for explicit occlusion modeling. Future models could incorporate temporal consistency constraints (Feichtenhofer et al., 2019) or amodal completion networks (Qi et al., 2021) to reason about partially visible objects. **Fine-Grained Semantic Enhancement.** To overcome CLIP's limitations in fine-grained categorization, future work could integrate domain-specific knowledge bases (Marino et al., 2019) or leverage large language models (OpenAI, 2023) to enrich semantic priors with detailed attribute descriptions. **Generalized Multi-modal Pretraining.** Extending our approach to unseen modalities like LiDAR and radar requires developing modality-agnostic fusion strategies. Unified multi-modal transformers (Akbari et al., 2021) pretrained on diverse sensor data could enhance generalization under domain shift. **Unsupervised Domain Adaptation.** To mitigate training complexity, future work should investigate self-supervised domain adaptation techniques (Ganin et al., 2016) that reduce the need for carefully designed curricula and extensive hyperparameter tuning.

### A.11 ALGORITHM PSEUDOCODE

The detailed procedural steps of the algorithm introduced in Section 3 are summarized in the pseudocode of Algorithm 1, Algorithm 2 capturing the core computational process of our approach.

---

**Algorithm 1** Stage 1: RGB-Infrared Fusion for Robust Detection

---

1: **Input:** RGB image $I_{rgb}$, Infrared image $I_{ir}$
2: **Output:** Enhanced multi-modal features $\mathcal{F}_{enhanced}$
3: **// Dual-Modal Feature Extraction**
4: $\mathcal{F}_{rgb} = \text{ResNet50}_{vis}(I_{rgb})$
5: $\mathcal{F}_{ir} = \text{ResNet50}_{ir}(I_{ir})$
6: **// RWKV-based Cross-Modal Fusion (VRWKV-Block)**
7: **for** each scale level $l \in \{2, 3, 4\}$ **do**
8:     **// Spatial Mix Module**
9:     $\mathcal{F}^l_{input} = \text{Concat}(\mathcal{F}^l_{rgb}, \mathcal{F}^l_{ir})$
10:     **// Q-Shift Operation**
11:     $\mathcal{F}^l_{shifted\_R} = \text{Q-Shift}_R(\mathcal{F}^l_{input})$
12:     $\mathcal{F}^l_{shifted\_K} = \text{Q-Shift}_K(\mathcal{F}^l_{input})$
13:     $\mathcal{F}^l_{shifted\_V} = \text{Q-Shift}_V(\mathcal{F}^l_{input})$
14:     **// Linear Projections**
15:     $R^l_s = \mathcal{F}^l_{shifted\_R}\mathbf{W}_R, K^l_s = \mathcal{F}^l_{shifted\_K}\mathbf{W}_K, V^l_s = \mathcal{F}^l_{shifted\_V}\mathbf{W}_V$
16:     **// BiWKV Attention**
17:     $\text{wkv}^l = \text{BiWKV}(K^l_s, V^l_s)$ using Eq. 2
18:     **// Spatial Output**
19:     $O^l_s = (\sigma(R^l_s) \odot \text{wkv}^l)\mathbf{W}_O$
20:     $\mathcal{F}^l_{spatial} = \text{LayerNorm}(O^l_s + \mathcal{F}^l_{input})$
21:     **// Channel Mix Module**
22:     $\mathcal{F}^l_{shifted\_R\_c} = \text{Q-Shift}_R(\mathcal{F}^l_{spatial})$
23:     $\mathcal{F}^l_{shifted\_K\_c} = \text{Q-Shift}_K(\mathcal{F}^l_{spatial})$
24:     $R^l_c = \mathcal{F}^l_{shifted\_R\_c}\mathbf{W}_{R\_c}, K^l_c = \mathcal{F}^l_{shifted\_K\_c}\mathbf{W}_{K\_c}$
25:     $V^l_c = \text{SquaredReLU}(K^l_c)\mathbf{W}_{V\_c}$
26:     $O^l_c = (\sigma(R^l_c) \odot V^l_c)\mathbf{W}_{O\_c}$
27:     $\mathcal{F}^l_{fused} = \text{LayerNorm}(O^l_c + \mathcal{F}^l_{spatial})$
28: **end for**
29: **// Feature Enhancement**
30: $\mathcal{F}_{enhanced} = \text{FPN}(\{\mathcal{F}^2_{fused}, \mathcal{F}^3_{fused}, \mathcal{F}^4_{fused}\})$
31: **// Detection Forward**
32: **if** training mode **then**
33:     $\mathcal{L}_{det} = \text{Query-based Head}(\mathcal{F}_{enhanced}, \text{gt\_bboxes}, \text{gt\_labels})$
34: **else**
35:     $\text{detections} = \text{Query-based Head.Test}(\mathcal{F}_{enhanced})$
36: **end if**
37: **return** $\mathcal{F}_{enhanced}, \mathcal{L}_{det}$

---

## A.12 SOCIAL IMPACT

Our method points to a promising approach for Open World Detection, significantly enhancing the reliability of vision systems in safety-critical applications. By leveraging robust multi-modal (RGB-Thermal) fusion, our framework demonstrates superior performance in challenging conditions such as low illumination, fog, and occlusion. This capability is crucial for enabling all-weather autonomous driving systems, ensuring safer navigation where traditional vision fails. Furthermore, it empowers Unmanned Aerial Vehicles (UAVs) for more effective search-and-rescue and disaster monitoring missions in adverse environments. The integration of open-vocabulary detection also allows these systems to dynamically recognize novel objects, expanding their utility in real-world scenarios without the need for costly re-training. We believe our work paves the way for more adaptable and trustworthy AI systems that can operate reliably in the open world, ultimately contributing to enhanced public safety and operational efficiency in fields like transportation, surveillance, and emergency response.

**Algorithm 2** Stage 2: Vision-Language Generalization Alignment

**Require:** Visual features $\mathbf{C} = \{C_2, C_3, C_4\}$ from backbone
**Require:** Text embeddings $\mathbf{T_{clip}}$ from CLIP model
**Require:** Ground truth boxes and labels
**Ensure:** Aligned vision-language features for detection

1: **// Semantic Enhancement Fusion**
2: $C_i' \leftarrow \text{Linear}(C_i), \quad i \in \{2, 3, 4\}$
3: $\mathbf{C} \leftarrow \text{Concat}(C_2', C_3', C_4')$
4: $\mathbf{M}, (H_p, W_p) \leftarrow \text{PatchEmbed}(C_2)$
5: **for** $R = 1$ to $l$ **do**
6: $\quad \mathbf{V}_I \leftarrow \text{RWKV}_{I \rightarrow T}(R = \mathbf{M}, K = \mathbf{T_{clip}}, V = \mathbf{T_{clip}})$
7: $\quad \mathbf{V}_T \leftarrow \text{RWKV}_{T \rightarrow I}(R = \mathbf{T_{clip}}, K = \mathbf{M}, V = \mathbf{M})$
8: $\quad \gamma \leftarrow \text{MLP}([\mathbf{M} \oplus \mathbf{V}_I])$
9: $\quad \delta \leftarrow \text{MLP}([\mathbf{T_{clip}} \oplus \mathbf{V}_T])$
10: $\quad \mathbf{M}_{out} \leftarrow \mathbf{M} + \gamma \otimes \mathbf{V}_I$
11: $\quad \mathbf{T}_{out} \leftarrow \mathbf{T_{clip}} + \delta \otimes \mathbf{V}_T$
12: **end for**
13:
14: **// Text-Modulated Sampling**
15: $\Delta_{base} \leftarrow \text{Linear}(\mathbf{Q})$
16: $\mathbf{A} \leftarrow \text{Softmax}(\mathbf{Q} \cdot \mathbf{T}_{text}^T / \sqrt{D})$
17: $\mathbf{W}_{mod} \leftarrow \text{MLP}(\mathbf{A})$
18: $\Delta_{updated} \leftarrow \Delta_{base} \oplus \mathbf{W}_{mod}$
19: $\mathbf{F}_{sampled} \leftarrow \text{Sample}(\mathbf{F}_{ref}, \mathbf{p}_{ref} + \Delta_{updated})$
20:
21: **// Bi-Momentum Contrastive Alignment**
22: **Initialize:** Queue $\mathcal{Q}_{region} \in \mathbb{R}^{K \times D_{proj}}$, Queue $\mathcal{Q}_{text} \in \mathbb{R}^{K \times D_{proj}}$
23: $\mathcal{P}_{pos} \leftarrow \{p_i | \text{IoU}(p_i, g_j) \geq \tau_{IoU}\}$
24: **for** each positive proposal $p \in \mathcal{P}_{pos}$ **do**
25: $\quad \mathbf{r}_q \leftarrow f_\theta(\text{RoIExtractor}(\mathbf{F}_{out}, p))$
26: $\quad \mathbf{r}_k \leftarrow f_{\theta_m}(\text{RoIExtractor}(\mathbf{F}_{out}, p))$
27: **end for**
28: **// Concatenate with queue negatives**
29: $\mathbf{K}_{text} \leftarrow \text{Concat}([\mathbf{t}_k, \mathcal{Q}_{text}])$
30: $\mathbf{K}_{region} \leftarrow \text{Concat}([\mathbf{r}_k, \mathcal{Q}_{region}])$
31: **// Multi-Positive InfoNCE Loss with queue negatives**
32: $s_{ij} = \langle \mathbf{r}_{q,i}, \mathbf{K}_{text,j} \rangle \cdot \exp(\alpha)$
33: $\mathcal{L}_{i2t} \leftarrow -\frac{1}{N_r} \sum_i \log \frac{\sum_{j \in \mathcal{P}_i^+} \exp(s_{ij}/\tau)}{\sum_{j=1}^{N_t + K} \exp(s_{ij}/\tau)}$
34: $\mathcal{L}_{t2i} \leftarrow -\frac{1}{N_t} \sum_j \log \frac{\sum_{i \in \mathcal{P}_j^+} \exp(s_{ji}/\tau)}{\sum_{i=1}^{N_r + K} \exp(s_{ji}/\tau)}$
35: $\mathcal{L}_{contrast} \leftarrow \lambda_{i2t} \mathcal{L}_{i2t} + \lambda_{t2i} \mathcal{L}_{t2i}$
36: **// Update momentum encoder via EMA**
37: $\theta_m \leftarrow m \cdot \theta_m + (1 - m) \cdot \theta$
38: **// Update queues**
39: $\mathcal{Q}_{region}.\text{dequeue}()$
40: $\mathcal{Q}_{region}.\text{enqueue}(\mathbf{r}_k)$
41: $\mathcal{Q}_{text}.\text{dequeue}()$
42: $\mathcal{Q}_{text}.\text{enqueue}(\mathbf{t}_k)$
43: **// Combine with detection loss**
44: $\mathcal{L}_{det} + \lambda_c \mathcal{L}_{contrast} + \lambda_{aux} \mathcal{L}_{aux}.$
45:
46: **return** Results, Detect loss $\mathcal{L}_{det}$

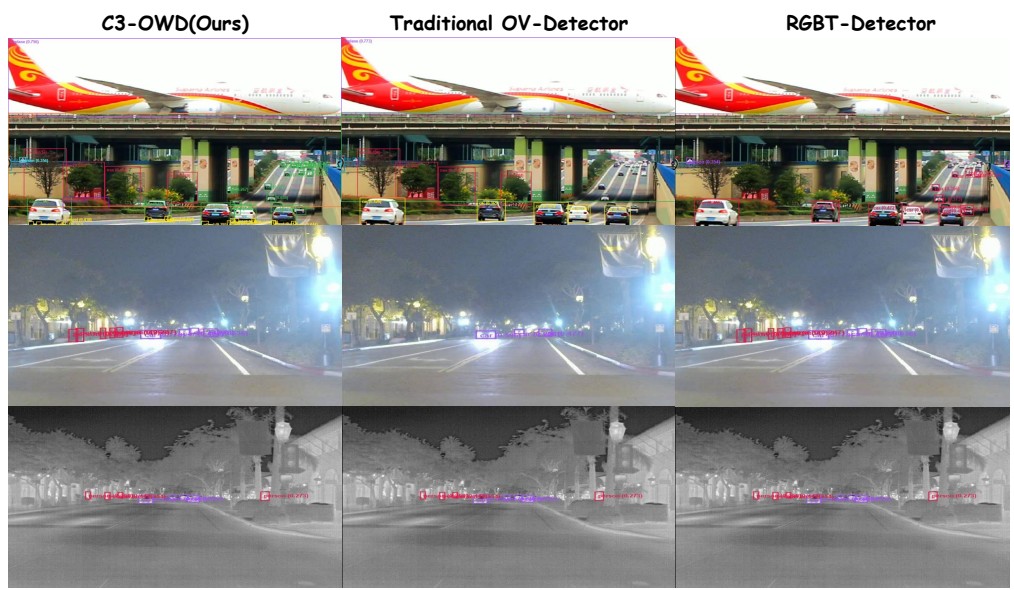

Figure 3: A Comparative Evaluation of Our Model against Traditional OVOD and Traditional RGBT Detection Models.

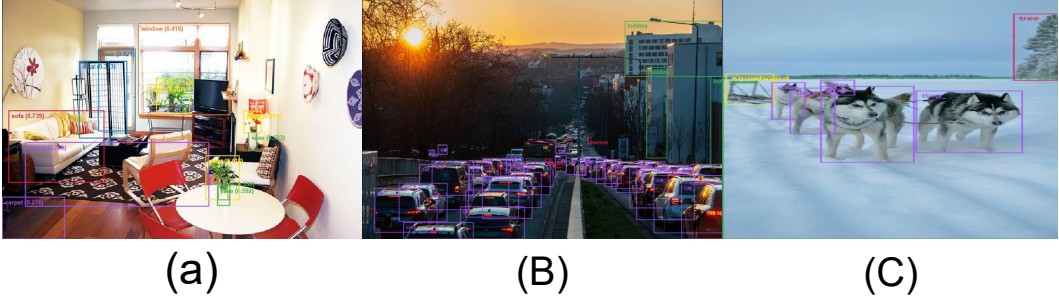

Figure 4: Representative examples of failure cases.

