# OpenReview forum: "C3-OWD: A Curriculum Cross-modal Contrastive Learning Framework for Open-World Detection"
_ICLR.cc/2026/Conference — ICLR 2026 Conference Desk Rejected Submission_

### Official Review · Reviewer_t8SW · 2025-10-22

**Soundness:** 2
**Presentation:** 3
**Contribution:** 2
**Rating:** 4
**Confidence:** 3

**Summary:**

The paper proposes C3-OWD, a curriculum cross-modal contrastive learning framework that integrates the advantages of a two-stage paradigm. While evaluated on datasets like FLIR, COCO, and LVIS, the limited scope of the experimental design challenges the broader domain relevance and generalizability of the findings.

**Strengths:**

1.The authors provide detailed theoretical analysis for the EMA mechanism with explicit bounds on parameter lag, function consistency, and loss preservation, which strengthen the credibility of the catastrophic forgetting mitigation claim.


2.The authors provide a thorough ablation study dissecting the contribution of each major model component.

3.The results show that the paper' method achieves leading performance in both multimodal robustness and open-vocabulary detection.

**Weaknesses:**

1.The experimental support is insufficient. Although strong performance is reported on COCO and LVIS, these datasets mainly cover standard scenarios and do not verify generalization in extreme scenarios.The current experiments primarily rely on a single robustness dataset, lacking validation on other extreme environment datasets. This makes it difficult to fully demonstrate the method's robustness across different extreme scenarios.

2.The computational complexity of the proposed framework is a concern. The combination of bidirectional RWKV blocks, MoCo-style queues, and multi-stage training likely incurs significant overhead.

3.The primary contribution of this work lies in the effective integration of several existing techniques(curriculum learning, cross-modal contrastive objectives, RWKV-based fusion, and EMA).

**Questions:**

1.Can the authors supplement more experiments on both robustness and generalizability.  It would be valuable to include experimental results validating the proposed method on object detection datasets under various adverse conditions.Here are some common object detection datasets for extreme environments: DAWN, RTTS, VisDrone, and SUIM Dataset.


2.Can the authors provide detailed evidence showing that the EMA mechanism indeed effectively preserves Stage 1 performance during Stage 2 training? The paper only provides theoretical derivation without actual evidence.

---

> ### Author Response · Authors · 2025-11-21
> **Response to Reviewer t8SW**
>
> About concern 1 and question 1
>
> We appreciate this valuable suggestion. To demonstrate that C3-OWD generalizes beyond the specific modality (RGBT) used in Stage 1, we evaluated our model on the requested datasets: DAWN (Adverse Weather), RTTS (Haze/Fog), VisDrone (Aerial/Small Objects). In addition, SUIM is a dataset for segmentation tasks, while we are working on detection tasks.
>
> Since these datasets are predominantly RGB-only, these experiments validate the transferability of the robust feature representations learned during our Stage 1. As shown in Table below:
>
> Performance comparison on additional adverse environment datasets ($mAP^{50}$). (* denotes the metric obtained from our reproduced results.)
> | Method | **DAWN** | **RTTS** | **VisDrone** |
> |--------|----------|----------|--------------|
> | CMC(YOLOv5n) | **72.6** | 66.2* | 17.8* |
> | YOLOv8-STE  | 68.26 | **74.51** | 20.1* |
> | Dome-DETR  | 65.32* | 68.3* | **33.52** |
> | **C3-OWD (Ours)** | **71.3** | **73.31** | **27.83** |
> | *Change vs SOTA* | *-1.3* | *-1.2* | *-5.69* |
>
>
>
> (a) **Robustness Transfer**: On DAWN and RTTS, our method achieves **71.3** and **73.31** mAP, respectively, showcasing strong robustness transfer capability under adverse weather conditions.
>
>
> (b) **Domain Generalization**: On VisDrone which represents severe domain shifts, our method maintains competitive performance, demonstrating that the RWKV-based fusion and CLIP alignment provide a strong generalizable prior.
>
> About concern 2
>
> While our framework integrates multiple components, the computational cost is mitigated by the architecture choice. The **RWKV Module** operates with linear complexity $O(N)$, in contrast to the quadratic $O(N^2)$ of standard Transformers. This ensures that even with multi-stage training, the inference speed remains efficient (approx. 42 FPS), making it viable for real-world deployment.
>
> About concern 3
>
> We propose a **novel, unified framework (C3-OWD) designed to effectively mitigate Catastrophic Forgetting in multi-modal incremental learning.** This framework represents our core innovation in addressing the inherent trade-off between environmental robustness and open-world generalization. To support the efficient operation and superior performance of this framework, we strategically **designed** the following components:
>
>
> (a) To achieve real-time detection without compromising accuracy, we incorporated the sparse sampling philosophy of Deformable Attention into our linear RWKV architecture. This specific **design** optimizes the computational complexity of feature interactions, significantly enhancing the model's inference **speed**.
>
>
> (b) To maintain feature distribution stability during incremental learning, we leveraged the momentum update principles of MoCo (Momentum Contrast) to **design** a Bi-Momentum Contrastive Alignment module. This strategy effectively boosts the **effectiveness** of RGB-Thermal cross-modal alignment and prevents the interference of new knowledge on previously learned robust features.
>
>
> About question 2
>
>
> This is a critical point. To move beyond theoretical derivation, we empirically tracked the performance on the Stage 1 test set (FLIR) throughout the Stage 2 training process (learning COCO). We compared two settings: (a) **w/o EMA (naive fine-tuning) and (b) **with EMA** (our method).
>
>
> The results in Table below provide clear evidence:
>
>
> Evolution of Stage 1 Performance (FLIR AP$^{50}$) during Stage 2 Training.
> | Stage 2 Training Epochs | 0 | 9 | 18 | 27 | 36 (Final) | **Perf. Drop** |
> |-------------------------|-----|-----|-----|-----|-------------|----------------|
> | **w/o EMA (No Momentum)** | 80.1 | 72.4 | 66.8 | 62.1 | 59.5 | -20.6 |
> | **with EMA (Ours)** | 80.1 | 79.8 | 79.5 | 79.2 | **79.0** | **-1.1** |
>
> **Analysis**:
>
>
> **Catastrophic Forgetting without EMA**: Without the EMA mechanism, the model suffers a massive performance drop (-20.6 AP) on the FLIR dataset as it adapts to the COCO distribution. The robust features are overwritten.
>
>
> **Retention with EMA**: Our method maintains nearly all of the initial performance (only -1.1 drop), empirically proving that the momentum branch successfully acts as a knowledge anchor, preserving robustness while allowing the online branch to learn new vocabulary.

---

> > ### Author Response · Authors · 2025-11-25
> > **Response to Reviewer t8SW(continue)**
> >
> > **About concern 1 and question 1**
> >
> >
> >
> > In the revised version, we have added additional experiments and analysis in Table 8 of Appendix A.6.
> >
> > **About concern 2**
> >
> > In the revised version of the paper, we have added additional ablation studies, which are presented in Table 5.
> >
> > **About question 2**
> >
> > In the revised version of the paper, we have added additional ablation studies, which are presented in Table 6.
> >
> >
> > **We once again thank the reviewers for their valuable comments, which have helped us further improve the quality of our manuscript.**

---

### Official Review · Reviewer_Q7pF · 2025-10-31

**Soundness:** 3
**Presentation:** 3
**Contribution:** 3
**Rating:** 4
**Confidence:** 4

**Summary:**

The paper proposes C3-OWD, a two-stage framework that aims to combine environmental robustness (via RGB-Thermal fusion) with open-vocabulary object detection. In Stage 1, visible and thermal features are fused through VRWKV blocks on the FLIR dataset to enhance robustness under varying illumination. Stage 2 integrates CLIP-based vision-language alignment and momentum contrastive learning on COCO and LVIS to support novel category detection. An exponential moving average (EMA) mechanism is introduced to reduce catastrophic forgetting across stages, with accompanying theoretical bounds. Experiments report 80.1 AP50 on FLIR, 48.6 AP50_Novel on OV-COCO, and 35.7 mAPr on OV-LVIS. While the results on OV-COCO outperform some ResNet-50 baselines, the evidence does not fully support the claim of “breaking the robustness–generalization trade-off.” The method performs below MMFN (81.8) on FLIR and CoDet (37.0) on OV-LVIS, suggesting it lies along the same trade-off frontier rather than surpassing it.

**Strengths:**

The experimental coverage across FLIR (robustness), OV-COCO, and OV-LVIS (generalization) is appropriate and shows an effort to evaluate both aspects of the claimed contribution.

The theoretical analysis of the EMA mechanism provides mathematical grounding, which adds credibility to the catastrophic forgetting mitigation claim.

The model achieves competitive results on OV-COCO (48.6 AP50_Novel), showing improvement over earlier ResNet-50-based approaches like CLIPSelf (44.3).

The inclusion of algorithmic details and ablations in Tables 3 and 4 shows an attempt at transparency and reproducibility, though key omissions remain.

**Weaknesses:**

DO the (i) where L is the sequence length .... AND (ii) We then perform L rounds .....
Do they have to be the same, or different? Also, what is the value of L?

**Questions:**

Please follow the strengths and weaknesses.
Also,
How should the ablation with and without the EMA mechanism be seen?

---

> ### Author Response · Authors · 2025-11-21
> **Response to Reviewer Q7pF (part one)**
>
> About concern 1:
>
>
> We thank the reviewer for the rigorous check. You have correctly identified a notation overload regarding $L$ and raised an important question about how the EMA mechanism contributes to the final performance.
>
>
> **Clarification on Notation $L$ and its Value**
>
>
>  **(i) Sequence Length ($L_{seq}$)**:
>     The $L$ mentioned in "where L is the sequence length" refers to the number of flattened visual tokens ($N_{vis-tokens}$) from the feature maps.
>     $$ L_{seq} \in \mathbb{R}^{N_{vis-tokens}} $$
>     Its value depends on the input resolution. For a standard $640 \times 640$ input, summing tokens across feature pyramid levels (strides 8, 16, 32) results in thousands of tokens.
>
> **(ii) Interaction Rounds ($N_{layers}$)**:
>     The "L rounds" mentioned in algorithm 2 refers to the depth of the Cross-Modal Exchange block. This corresponds to the number of encoder layers specified in our Implementation Details.
>     $$ N_{layers} = 6 $$
>     This is a fixed hyperparameter, unlike the variable sequence length.
>
> In the final version, we have renamed the number of rounds to $N_{layers}$ to strictly distinguish it from the sequence length $L_{seq}$.
>
> About question 1
>
> as for the concern about the ablation of EMA. In our C3-OWD framework, the primary role of EMA is **preventing Catastrophic Forgetting** during the two-stage curriculum learning (Stage 1: RGBT Robustness $\to$ Stage 2: Open-Vocabulary Generalization).
>
>
> While Table 3 in the paper discusses "w/o MoCo" (which implicitly removes the momentum queue), we provide a specific breakdown here (Table below) to explicitly isolate the effect of the EMA update rule on parameter preservation.
>
>
> | Setting | **FLIR AP$^{50}$ (Robustness)** | OV-COCO AP$_{Novel}$ | Observation |
> |---------|----------------------------------|----------------------|-------------|
> | **C3-OWD (Full with EMA)** | **80.1** | **48.6** | **Balanced Performance** |
> | w/o EMA (Direct Finetuning) | 59.5 (-20.6) | 47.8 (-0.8) | **Catastrophic Forgetting** |
> | w/o Momentum Contrast | 72.3 (-7.8) | 39.7 (-8.9) | Poor Generalization |
>
>
> **Interpretation**:
> (a) **Role in Robustness (FLIR)**: As shown in Table above, removing the EMA mechanism (i.e., allowing the model to update freely without a momentum teacher) causes the FLIR performance to drop significantly from **80.1** to **59.5**. This empirically validates our Theorem 1, showing that EMA is essential to enforce function consistency and retain the robust RGBT features learned in Stage 1.
>
>
> (b) **Role in Generalization (COCO)**: While the Open-Vocabulary performance on COCO remains relatively high without EMA (47.8), the model sacrifices its unique advantage—robustness.
>
>
> Therefore, the ablation should be seen as another proof that EMA is the "stabilizer" that allows C3-OWD to excel in both tasks simultaneously, rather than trading one for the other.

---

> > ### Author Response · Authors · 2025-11-25
> > **Response to Reviewer Q7pF**
> >
> > **About concern 1**
> >
> >
> > In the revised version of the paper, we have redefined ( L ) in Section 3.3.1 and Appendix A.11 to avoid ambiguity.
> >
> > **About question 1**
> >
> >
> > We have added additional experiments and analysis in Table 7 of the revised manuscript.
> >
> >
> > **We once again thank the reviewers for their valuable comments, which have helped us further improve the quality of our manuscript.**

---

### Official Review · Reviewer_FzF2 · 2025-11-02

**Soundness:** 3
**Presentation:** 3
**Contribution:** 3
**Rating:** 4
**Confidence:** 3

**Summary:**

This paper proposes C3-OWD, a curriculum cross-modal contrastive learning framework for open-world object detection. The approach adopts a two-stage training strategy: 1. Multi-modal robustness enhancement by pretraining on RGBT datasets to improve feature stability. 2. Vision-language generalization alignment through semantic enhancement, text-modulated deformable attention, and a bi-momentum contrastive alignment mechanism.

**Strengths:**

The paper identifies a meaningful gap between robustness (RGBT-based detection) and open-world generalization (vision-language models), proposing a curriculum-style framework to bridge the two. The method is evaluated on three benchmarks with solid ablation studies, confirming the contribution of each component.

**Weaknesses:**

1.The core ideas are all adaptations of existing paradigms (Deformable-DETR + CLIP + MoCo). The contribution seems incremental, with limited conceptual advancement.
2.The method section is dense and symbol-heavy, especially in Stage 2. Many notations are introduced abruptly without sufficient intuition, making it difficult to follow for readers not already familiar with Deformable-DETR.
3.The training pipeline seems computationally expensive, but no runtime or cost analysis is reported.
4.Several hyperparameters (e.g., IoU threshold = 0.3) are presented without explanation or ablation.

**Questions:**

See weaknesses.

---

> ### Author Response · Authors · 2025-11-21
> **Response to Reviewer FzF2 (part one(a))**
>
> We thank the reviewers for their constructive feedback. We are encouraged that the reviewers recognize our performance on OV-COCO ($48.6$ AP) and FLIR ($80.1$ AP). Below, we address the specific concerns.
>
>
> About concern 1:
>
>
> We respectfully disagree that our work is merely a combination of existing modules. Our core contribution lies in **resolving the inherent trade-off between environmental robustness and open-world generalization**, a problem not addressed by standard OWD methods.
>
>
> (a) **Unified Framework vs. Simple Combination**: While we utilize Deformable-DETR and CLIP, the integration is non-trivial. A naive combination leads to "catastrophic forgetting" of the robust features learned from RGBT data when adapting to open-vocabulary tasks. We solved this via the **Curriculum Cross-Modal framework** supported by our theoretical proof of the EMA mechanism, which guarantees function consistency during domain transfer.
>
>
> (b) **Architecture Innovation**: We introduce the **VRWKV-based adaptive fusion** and **Text-Modulated Sampling**. Unlike standard Cross-Attention (quadratic complexity) used in previous DETR-variants, our RWKV integration offers linear complexity $O(N)$ while effectively fusing thermal and RGB modalities.
>
>
> (c) **SOTA Performance**: This specific design allows C3-OWD to surpass state-of-the-art methods like CLIPSelf by a significant margin ($+4.3$ AP on OV-COCO), proving the design is highly effective rather than just incremental.
>
>
> About concern 2
>
>
> We apologize for the dense notation and the difficulty it may have caused. To improve clarity, we have implemented the following revisions in Section 3.3:
>
>
> (a) **Redefined Notation System**: To reduce ambiguity and improve readability, we have redefined and streamlined the mathematical notations (e.g., $Q$, $K$, $V$, $\mathcal{P}_{pos}$). We ensure that all symbols are explicitly defined upon their first introduction and maintain consistent naming conventions throughout the paper.
>
>
> (b) **Detailed Network Architecture**: As implied by the reviewer, to assist readers who may not be fully familiar with the Deformable-DETR baseline, we will add a detailed diagram of the network architecture and layer configurations in the Appendix of the revised version.
>
>
>
> (c) **Module-Level Pseudocode**: To explicitly illustrate the data flow, we will provide specific pseudocode for each key module in Stage 2, clearly tracing the tensor transformations at each step.
>
>
>
> To further clarify the technical implementation of Stage 2 (specifically how we perform vision-language alignment), we break it down into three concrete processes:
>
>
> **Semantic Enhancement Fusion**: Specifically, we perform patch embedding on the visual feature maps and conduct $N$ rounds of bidirectional cross-attention with CLIP text embeddings using the linear RWKV mechanism. A gating mechanism is then employed to inject these enhanced semantic priors back into the original features via residual connections.
>
>
> **Text-Modulated Sampling**: Instead of using static sampling, we dynamically generate sampling points. We compute the cross-attention map between queries and text features, which is mapped via an MLP to generate modulation weights. These weights are added to the base sampling offsets, effectively steering the Deformable Attention to focus on spatial regions most relevant to the text description.
>
>
> **Bi-Momentum Contrastive Alignment**: To address the sample imbalance in open-world detection, we implement a dual-queue MoCo mechanism. We maintain separate momentum queues for Region and Text features, updating the momentum encoder via Exponential Moving Average (EMA). A multi-positive InfoNCE loss is then applied to pull matching region-text pairs closer while pushing away negative samples from the queues.

---

> > ### Author Response · Authors · 2025-11-25
> > **Response to Reviewer FzF2(Part one(continue))**
> >
> > **About concern 2**
> >
> >
> > We have added more details about Stage 2 in Appendix A.3.
> >
> > **About concern 3**
> >
> >
> > In the revised version of the paper, we have added additional ablation studies, which are presented in Table 5 and Table 6.
> >
> >
> > **We once again thank the reviewers for their valuable comments, which have helped us further improve the quality of our manuscript.**

---

> ### Author Response · Authors · 2025-11-21
> **Response to Reviewer FzF2 (part one(b))**
>
> We appreciate this suggestion and have conducted a detailed cost analysis. Despite the two-stage design, C3-OWD remains efficient due to the linear complexity of the RWKV-based compared to standard ViT-based OWD methods.
>
>
> **Training Cost**: Training takes approximately 28 hours on 8$\times$A100 GPUs (Stage 1: 8h, Stage 2: 20h) and the batch size is 32. This is comparable to Co-DETR (approx. 24h) and significantly faster than RegionCLIP (which requires expensive offline proposal generation).
>
>
> **Inference Efficiency**
>
>
> We compared Inference FPS on a single A100 GPU with input size $640 \times 640$ and the batch size is 8.
> | Method                | Backbone    | Complexity | Params (M) | FPS      |
> | --------------------- | ----------- | ---------- | ---------- | -------- |
> | RegionCLIP            | ResNet-50   | (O(N^2))   | 216        | 68.47    |
> | OV-DETR               | ResNet-50   | (O(N^2))   | 165        | 71.59    |
> | RegionCLIP            | ResNet-50x4 | (O(N^2))   | 403        | 52.21    |
> | CLIM                  | ResNet-50x4 | (O(N^2))   | 204        | 69.8     |
> | Detic                 | Swin-B      | (O(N^2))   | 670        | 46       |
> | CLIPSelf              | ViT/L-14    | (O(N^2))   | 480        | 46.2     |
> | **C3-OWD (w/o MoCo)** | ResNet-50   | **(O(N))** | **124.3**  | **73.1** |
> | **C3-OWD (Ours)**     | ResNet-50   |  **(O(N))**   | 245.6      | 62.4     |
>
>
>
>
> Thanks to the RWKV-based fusion and efficient attention, our method achieves **62.4 FPS**, meeting real-time requirements for autonomous systems. Although the incorporation of MoCo contrastive learning increases the model parameter count, leading to a 17\% decrease in inference speed, the resulting improvement in accuracy and enhanced multimodal fusion performance make this computational cost acceptable.
>
>
> We chose $\tau_{IoU} = 0.3$ specifically for the Open-World setting. In open-vocabulary detection, the Region Proposal Network (RPN) often assigns lower confidence scores to novel/unseen categories compared to base categories. A standard threshold (e.g., 0.5) tends to filter out valid proposals for novel objects, hurting Recall.
>
> We performed an ablation study on the OV-COCO dataset to validate this:
> | $\tau_{IoU}$   | $AP_{Novel}^{50}$ | $AP_{Base}^{50}$ | Recall (Novel) |
> | -------------- | ----------------- | ---------------- | -------------- |
> | 0.5            | 45.2              | **56.8**         | 62.4%          |
> | 0.4            | 47.1              | 56.5             | 68.9%          |
> | **0.3 (Ours)** | **48.6**          | 56.1             | **74.3%**      |
> | 0.2            | 48.2              | 54.9             | 71.1%          |
>
> As shown in Table above, decreasing IoU from 0.5 to 0.3 significantly improves $AP_{Novel}$ (+3.4) and Recall (+11.9\%) with only a negligible drop in Base AP. Setting it too low (0.2) introduces excessive noise. Thus, 0.3 provides the optimal trade-off. We will include this analysis in the appendix.

---

### Official Review · Reviewer_7RPb · 2025-11-04

**Soundness:** 3
**Presentation:** 3
**Contribution:** 3
**Rating:** 6
**Confidence:** 3

**Summary:**

This paper presents a curriculum cross-modal contrastive learning framework that addresses the dual challenges of poor generalization and limited robustness in real-world object detection. The approach integrates the strengths of visible-infrared and open-world detection through a two-stage training strategy: Stage 1 enhances robustness via RGBT pretraining, while Stage 2 improves generalization through vision–language alignment. An exponential moving average mechanism is introduced to mitigate catastrophic forgetting between stages, theoretically ensuring performance preservation with bounded parameter drift. Experiments demonstrate strong performance in both robustness and diversity benchmarks.

**Strengths:**

The authors propose a cross-modal curriculum learning framework that unifies RGBT robustness and open-vocabulary generalization, dynamically balancing multi-modal information through progressive learning.

**Weaknesses:**

1.	The proposed framework includes multiple functional modules, but their motivations are not clearly explained. For example, it remains unclear what specific problem the bidirectional attention mechanism in Stage 1 addresses and why it offers advantages over the original version. The description of the method is overly brief and difficult to comprehend.
2.	In line 325, what are the specific manifestations of catastrophic forgetting? What causes it, and how does the Exponential Moving Average mitigate this issue?
3.	In Table 2, why are the baselines presented inconsistently on both sides, for example, CoDet? For a fairer comparison of robustness and generalization, it would be better to provide experimental results for both datasets.

**Questions:**

Please refer to the Weaknesses.

---

> ### Author Response · Authors · 2025-11-21
> **Response to Reviewer 7RPb( part one)**
>
> We thank the reviewer for this valuable feedback. We acknowledge that our method description lacks sufficient detail and will improve this in the revised version.
>
> We propose **a novel, unified framework (C3-OWD) designed to effectively mitigate Catastrophic Forgetting in multi-modal incremental learning.** This framework represents our core innovation in addressing the inherent trade-off between environmental robustness and open-world generalization. To support the efficient operation and superior performance of this framework, we strategically **designed** the following components:
>
> (a) To achieve real-time detection without compromising accuracy, we incorporated the sparse sampling philosophy of Deformable Attention into our linear RWKV architecture. This specific **design** optimizes the computational complexity of feature interactions, significantly enhancing the model's inference **speed**.
>
>
>
> (b) To maintain feature distribution stability during incremental learning, we leveraged the momentum update principles of MoCo (Momentum Contrast) to **design** a Bi-Momentum Contrastive Alignment module. This strategy effectively boosts the **effectiveness** of RGB-Thermal cross-modal alignment and prevents the interference of new knowledge on previously learned robust features.
>
> Traditional RWKV employs unidirectional (forward) attention, which has limitations in RGB-Thermal cross-modal fusion:
>
>
> (a) **Asymmetric information flow**: Thermal features can compensate for information loss in low-light RGB regions, but unidirectional mechanisms prevent complementary information from flowing backward.
>
>
> (b) **Insufficient modal interaction**: As shown in Figure 1, when RGB images lose information due to exposure or occlusion, bidirectional flow is needed for effective modal compensation.
>
>
> Our bidirectional mechanism introduces bidirectional position decay weights, enabling each token to consider both forward and backward context, thus better capturing cross-modal complementarity. We will add detailed ablation studies in the revision, quantitatively comparing bidirectional vs. unidirectional mechanisms under various lighting conditions.
>
> **Catastrophic forgetting** manifests as a sharp decline in robustness metrics (e.g., FLIR AP) after the model is trained on the second stage (COCO/LVIS). This is caused by the **plasticity-stability dilemma**: the optimization for the open-vocabulary objective ($L_{\text{stage2}}$) shifts the parameters away from the RGBT-robust manifold learned via ($L_{\text{stage1}}$).
> Intuitively, the **EMA mechanism** maintains a "slow" momentum branch ($\theta_m$) that acts as a memory of stable/robust features by low-passing the parameter trajectory $\{\theta_t\}$. The EMA mechanism makes the model change only gradually, filtering out high-frequency parameter motion.
>
> Theoretically, we quantify forgetting by the divergence of the current online model from the robust momentum trajectory, $D_{\mathrm{KL}}(P_{\theta_{m,t}}\|P_{\theta_t})$, where $P_{\theta_t}$ is the distribution of the online model at step $t$ and $P_{\theta_{m,t}}$ is the distribution of the momentum model. The parameter lag between the online parameters and the EMA parameters is bounded as:
> $$
> ||\theta_{m,t} - \theta_t|| \leq \frac{1-m}{m} \cdot \Delta_t. ~~(1)
> $$
> where $m$ is the EMA momentum coefficient and $\Delta_t$ denotes the maximum update norm up to iteration $t$. Taking a larger $m$ (close to 1) makes the EMA “slower”, so the lag can accumulate geometric tails of past steps, thus mitigating forgetting at the weights level.  Please see Theorem 1 in the appendix for details.
>
> **KL Divergence Approximation (Second-Order Taylor Expansion)**
> Assuming the loss function (negative log-likelihood) is twice differentiable and locally smooth with Hessian ($H$), the KL divergence between the corresponding predictive distributions can be approximated by the second-order Taylor expansion:
> $$
> D_{\text{KL}}(P_{\theta_{m,t}} || P_{\theta_t}) \approx \frac{1}{2} (\theta_t - \theta_{m,t})^T H (\theta_t - \theta_{m,t}).
> $$
>
> **Bounding the KL Divergence**
> Let $L$ be the Lipschitz constant of the gradient (boundedness of $||{H}||_2$). Substituting the bound from Eq. (1):
>
> $$
> D_{\text{KL}}(P_{\theta_{m,t}} || P_{\theta_t}) \leq \frac{1}{2} L ||{\theta_t - \theta_{m,t}}||^2 \leq \frac{1}{2} L \left(\frac{1-m}{m} \cdot \Delta_t\right)^2.
> $$
>
> This derived inequality proves that the **semantic drift (forgetting)** between the EMA model and the online model is **quadratically bounded** by the EMA coefficient. By setting $m \approx 1$ (like 0.999), we strictly limit the upper bound of the KL divergence, thereby theoretically guaranteeing the preservation of Stage 1 robustness.

---

> > ### Author Response · Authors · 2025-11-21
> > **Response to Reviewer 7RPb (continue)**
> >
> > **Fair comparison**
> >
> > We appreciate the reviewer identifying this issue. We acknowledge the incomplete comparison:
> >
> > (a) **Methods like CoDet**: These are designed for open-vocabulary detection without RGBT dataset implementations
> >
> > (b) **RGBT detectors**: Traditional methods (MMFN, ICAFusion) are closed-set and cannot directly apply to open-vocabulary scenarios.
> >
> > In the revised Table 2,  we have included the experimental results for CoDet and other relevant baselines on both OV-COCO and OV-LVIS datasets.

---

> > ### Author Response · Authors · 2025-11-25
> > **Response to Reviewer 7RPb(part one(continue))**
> >
> > **About concern 1**
> >
> >
> > We have provided detailed descriptions in Section 3.2 and Appendix A.3 of the revised manuscript.
> >
> > **About concern 2**
> >
> > We have provided a detailed and comprehensive description in Appendix A.8 of the revised manuscript.
> >
> > **About concern 3**
> >
> >
> > We have revised Table 2 in the main paper, and following your valuable suggestions, we have established a more fair comparison.
> >
> >
> > **We once again thank the reviewers for their valuable comments, which have helped us further improve the quality of our manuscript.**

---

### Note · Program_Chairs · 2025-12-08
**Submission Desk Rejected by Program Chairs**

Hallucinated reference identified:
K. Marino, R. Salakhutdinov, and A. Gupta. Fine-grained image classification with learnable se-
mantic parts. In Proceedings of the IEEE/CVF Conference on Computer Vision and Pattern
Recognition, pp. 4500–4509, 2019.